# Recruitment of TBK1 to cytosol-invading *Salmonella* induces WIPI2-dependent antibacterial autophagy

Teresa LM Thurston[1,2], Keith B Boyle[1], Mark Allen[3], Benjamin J Ravenhill[1], Maryia Karpiyevich[1], Stuart Bloor[1,†], Annie Kaul[1], Jessica Noad[1], Agnes Foeglein[1], Sophie A Matthews[2], David Komander[1], Mark Bycroft[3] & Felix Randow[1,4,*]

## Abstract

Mammalian cells deploy autophagy to defend their cytosol against bacterial invaders. Anti-bacterial autophagy relies on the core autophagy machinery, cargo receptors, and "eat-me" signals such as galectin-8 and ubiquitin that label bacteria as autophagy cargo. Anti-bacterial autophagy also requires the kinase TBK1, whose role in autophagy has remained enigmatic. Here we show that recruitment of WIPI2, itself essential for anti-bacterial autophagy, is dependent on the localization of catalytically active TBK1 to the vicinity of cytosolic bacteria. Experimental manipulation of TBK1 recruitment revealed that engagement of TBK1 with any of a variety of *Salmonella*-associated "eat-me" signals, including host-derived glycans and K48- and K63-linked ubiquitin chains, suffices to restrict bacterial proliferation. Promiscuity in recruiting TBK1 via independent signals may buffer TBK1 functionality from potential bacterial antagonism and thus be of evolutionary advantage to the host.

**Keywords** anti-bacterial autophagy; PI(3)P; *Salmonella*; TBK1; WIPI
**Subject Categories** Autophagy & Cell Death; Microbiology, Virology & Host Pathogen Interaction
The EMBO Journal (2016) 35: 1779–1792

## Introduction

Anti-bacterial autophagy provides potent cell-autonomous immunity against bacterial attempts to colonize the cytosol of mammalian cells (Kuballa *et al*, 2012; Deretic *et al*, 2013; Randow *et al*, 2013). The defense of the gut epithelium against bacteria in particular is crucially dependent on anti-bacterial autophagy, since mice lacking the essential autophagy gene *Atg5* in enterocytes suffer from tissue invasion by commensal bacteria and from increased pathology upon infection with *Salmonella enterica* serovar Typhimurium (*S.* Typhimurium), a specialized enteropathogen (Benjamin *et al*, 2013).

Macro-autophagy, hereafter autophagy, is an evolutionarily conserved quality control and degradation pathway that engulfs cytosolic content into double-membrane vesicles called autophagosomes. Autophagosome biogenesis requires the concerted activity of about 15 core AuTophaGy genes (ATGs), among them the VPS34 lipid kinase complex (Mizushima *et al*, 2011). VPS34 produces membrane patches rich in phosphatidylinositol 3-phosphate (PI(3)P) that recruit PI(3)P-binding proteins such as WIPI and DFCP1 to the site of phagophore formation (Axe *et al*, 2008). In contrast to the non-selective engulfment of cytosol into starvation-induced autophagosomes, anti-bacterial autophagy is mediated by cargo receptors including NDP52, optineurin, and p62 (Thurston *et al*, 2009; Zheng *et al*, 2009; Wild *et al*, 2011). Cargo receptors bind members of the LC3/GABARAP family of ubiquitin-like proteins on the autophagosomal membrane and specific "eat-me" signals associated with cytosol-invading bacteria, thereby selectively tethering bacteria to phagophore membranes (Weidberg *et al*, 2011; Rogov *et al*, 2014).

*Salmonella enterica* serovar Typhimurium reaches the cytosol from a vesicular compartment, the *Salmonella*-containing vacuole (SCV). Damage to the limiting membrane of the SCV during bacterial escape exposes host glycans otherwise hidden inside the vacuole as ligands for a family of cytosolic lectins, the galectins (Dupont *et al*, 2009; Paz *et al*, 2010). By binding the cargo receptor NDP52, galectin-8 provides an "eat-me" signal for anti-bacterial autophagy (Thurston *et al*, 2012). The dense layer of poly-ubiquitylated proteins that accumulates on cytosol-exposed *S.* Typhimurium serves as an alternative "eat-me" signal, which is sensed by multiple cargo receptors, namely NDP52, optineurin, and p62 (Perrin *et al*, 2004; Thurston *et al*, 2009; Zheng *et al*, 2009; Wild *et al*, 2011). Failure of "eat-me" signals to associate with cytosolic bacteria or interference with cargo receptor function prevents efficient anti-bacterial autophagy and allows hyper-proliferation of cytosolic *S.* Typhimurium (Boyle & Randow, 2013).

1 Division of Protein and Nucleic Acid Chemistry, MRC Laboratory of Molecular Biology, Cambridge, UK
2 MRC Centre for Molecular Bacteriology and Infection, Imperial College London, London, UK
3 Division of Structural Studies, MRC Laboratory of Molecular Biology, Cambridge, UK
4 Department of Medicine, Addenbrooke's Hospital, University of Cambridge, Cambridge, UK
*Corresponding author. Tel: +44 1223 267161; Fax: +44 1223 268306; E-mail: randow@mrc-lmb.cam.ac.uk
†Present address: Cambridge Institute for Medical Research, University of Cambridge, Cambridge, UK

Restricting the proliferation of *S.* Typhimurium also requires the kinase TBK1, a member of the IKK (inhibitor of nuclear factor κB kinase) family (Radtke *et al*, 2007; Thurston *et al*, 2009). The anti-bacterial function of TBK1 is distinct from its well-characterized role of inducing type I interferons by phosphorylating IRF3 in virally infected cells (Randow *et al*, 2013; Wu & Chen, 2014). TBK1 accumulates in the vicinity of cytosol-exposed bacteria together with its adaptor proteins Nap1, Sintbad, and their binding partner NDP52 (Fujita *et al*, 2003; Ryzhakov & Randow, 2007; Thurston *et al*, 2009; Verlhac *et al*, 2015). TBK1 also associates with optineurin and it has been reported to phosphorylate both optineurin and p62, thereby enhancing their affinity for LC3B and ubiquitin, respectively (Morton *et al*, 2008; Wild *et al*, 2011; Pilli *et al*, 2012; Heo *et al*, 2015; Richter *et al*, 2016). While these findings imply that TBK1 strengthens the tethering function of cargo receptors, TBK1 has also been suggested to promote autophagosome maturation (Pilli *et al*, 2012).

Here we show that in order to restrict *Salmonella* proliferation TBK1 activity is required in the proximity of cytosolic bacteria for the recruitment of WIPI2, a PI(3)P-binding upstream autophagy component itself essential for anti-bacterial autophagy. To investigate the recruitment requirements for TBK1 in restricting bacterial proliferation, we deployed a TBK1 variant unable to bind any of its known adaptors. Recruitment of TBK1 to *S.* Typhimurium via any of several eat-me signals, including galectin-8 and K48- or K63-linked ubiquitin, is sufficient to provide TBK1 functionality for anti-bacterial autophagy, suggesting that robust and promiscuous recruitment of TBK1 to cytosol-invading bacteria may be beneficial in thwarting potential bacterial evasion attempts.

# Results

## The autophagic capture of *Salmonella* requires enzymatically active TBK1 in the bacterial vicinity

TBK1 is essential for anti-bacterial autophagy but its precise function in the pathway, as well as its mode of activation, remain poorly understood. TBK1 comprises an N-terminal kinase domain, a ubiquitin-like domain, and two C-terminal coiled-coils. To explore the role of TBK1 in antagonizing *S.* Typhimurium replication inside host cells we utilized TBK1 knockout mouse embryonic fibroblasts (MEFs). We confirmed previous findings of unrestricted proliferation of *S.* Typhimurium in $Tbk1^{-/-}$ MEFs, a phenotype complemented with wild-type but not catalytically inactive $TBK1_{K38M}$ (Figs 1A and EV1A) (Pomerantz & Baltimore, 1999; Radtke *et al*, 2007). We have previously shown that TBK1 physically associates with those intracellular *Salmonella* that are positive for the TBK1 adaptor proteins Nap1 and Sintbad and the autophagy cargo receptor NDP52 (Thurston *et al*, 2009). To test whether the function of TBK1 in anti-bacterial autophagy requires interactions with its adaptor proteins we truncated TBK1 at its C-terminus ($TBK1_{N685}$ hereafter referred to as $TBK1_{\Delta C}$), thereby generating a molecule deficient in binding to all its known adaptors, that is Nap1, Sintbad, Tank, and optineurin (Fig EV1B) (Goncalves *et al*, 2011), while maintaining kinase activity as indicated by the activation of an ISRE reporter (Fig EV1C). Complementation of $Tbk1^{-/-}$ MEFS with $TBK1_{\Delta C}$ failed to restrict proliferation of *S.* Typhimurium (Figs 1A and EV1A). The

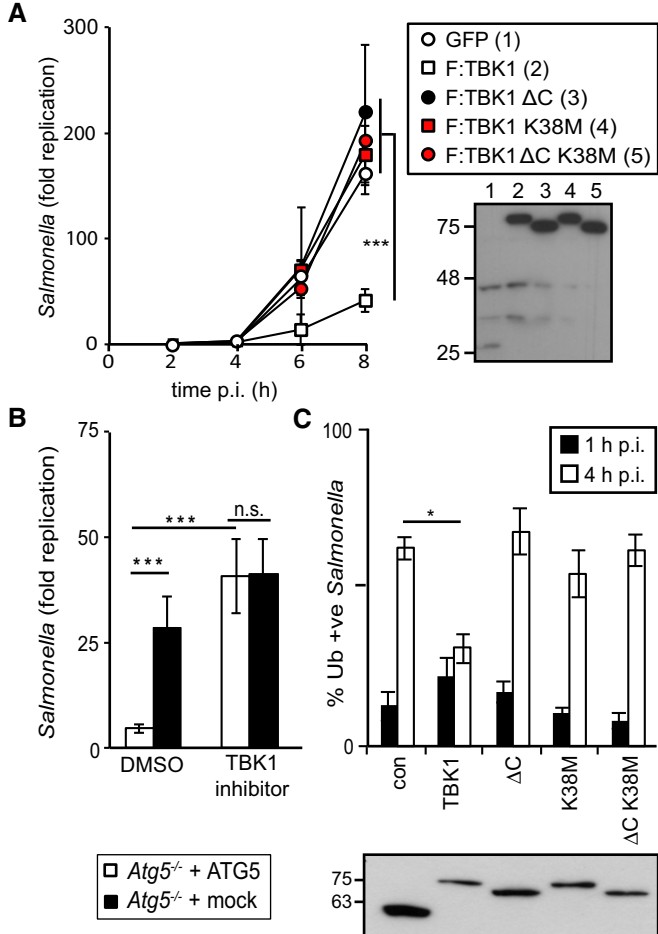

**Figure 1.  TBK1 kinase activity and C-terminal domain are required for restriction of *Salmonella enterica* serovar Typhimurium replication.**

A   Kinetics of *S.* Typhimurium replication in $Tbk1^{-/-}$ MEFs stably expressing the indicated TBK1 alleles. Intracellular bacteria were enumerated by their ability to form colonies on agar plates following cell lysis at the indicated time points. Statistical analysis comparing $Tbk1^{-/-}$ MEFs and cells expressing TBK1 alleles. Western blot for Flag-tagged TBK1 variants in post-nuclear cell lysates.

B   Replication of *S.* Typhimurium in $Atg5^{-/-}$ MEFs complemented with ATG5 or mock and treated with the TBK1 kinase inhibitor MRT68843 (10 nM) or DMSO. Fold replication was determined by counting bacterial colonies on agar plates at 2 and 8 h post-inoculation (p.i.) following cell lysis.

C   Analysis of $Tbk1^{-/-}$ MEFs stably expressing the indicated TBK1 alleles and infected with *S.* Typhimurium. Percentage of *S.* Typhimurium coated with ubiquitin (detected by FK2 antibody) at 1 or 4 h p.i. Western blot for Flag-tagged TBK1 variants in post-nuclear cell lysates.

Data information: Mean and SD of triplicate MEF cultures and duplicate colony counts. Data are representative of at least two repeats (A, B). Mean and SEM of three independent experiments with duplicate coverslips. > 200 bacteria counted per coverslip (C). *$P < 0.05$, ***$P < 0.001$, one-way ANOVA with Dunnett's multiple comparisons test.

double mutant lacking catalytic activity and adaptor binding had a phenotype no more severe than either single mutant in the *Salmonella* assay (Figs 1A and EV1A). We therefore conclude that the catalytic activity of TBK1 and its ability to bind adaptor proteins are equally important to protect cells against *S.* Typhimurium, most

likely because adaptor binding controls TBK1 spatially and/or temporally.

Precisely how TBK1 restricts bacterial proliferation is controversial; the enhanced bacterial load in TBK1-deficient cells has been suggested to be caused by either the cells' inability to maintain SCV integrity (Radtke *et al*, 2007) or the cells' inability to execute anti-bacterial autophagy (Thurston *et al*, 2009). The TBK1 inhibitor MRT68843, which inhibits poly(I:C)-induced ISRE reporter activity similar to the related TBK1 inhibitor MRT68601 (Newman *et al*, 2012) (Fig EV1D), was used to analyze the relationship between TBK1 activity and autophagy. As expected, $Atg5^{-/-}$ cells failed to suppress proliferation of *S.* Typhimurium (Fig 1B). Addition of MRT68843 increased bacterial replication only 1.5-fold in $Atg5^{-/-}$ MEFs but more than eightfold in cells complemented with ATG5, consistent with TBK1 controlling anti-bacterial autophagy due to its kinase activity. To substantiate this finding, we next investigated where in the anti-bacterial autophagy pathway TBK1 acts. By complementing $Tbk1^{-/-}$ MEFs, we confirmed that lack of TBK1 increased the percentage of ubiquitin-coated cytosolic *S.* Typhimurium at 4 h post-infection (Fig 1C) (Radtke *et al*, 2007; Thurston *et al*, 2009). $TBK1_{K38M}$ and $TBK1_{\Delta C}$, which are catalytically inactive and deficient in binding adaptor proteins, respectively, did not complement the ubiquitin phenotype, in line with the lack of these alleles to control proliferation of *S.* Typhimurium in the cytosol of host cells (Fig 1A and C).

## The recruitment of WIPI2, itself essential for anti-bacterial autophagy, is controlled by TBK1

The anti-bacterial autophagy attack can be visualized by assessing the association of *S.* Typhimurium with LC3B, a mammalian Atg8 ortholog. However, complementation of $Tbk1^{-/-}$ MEFs with wild-type TBK1 did not significantly alter the percentage of GFP: LC3B-positive *S.* Typhimurium at 1 h post-infection, nor did complementation with $TBK1_{\Delta C}$ or $TBK1_{K38M}$ (Fig 2A). Such apparently normal recruitment of LC3B to *S.* Typhimurium in cells failing to restrict bacterial proliferation (Fig 1A and C) may be due to conjugation of LC3 to the remnants of SCV membranes rather than anti-bacterial phagophores, a phenotype well-documented for MEFs with defects in upstream autophagy components such as FIP200 or ATG9 (Kageyama *et al*, 2011). The phenotype in $Tbk1^{-/-}$ MEFs therefore points to an upstream defect in the autophagy pathway. Phagophore formation requires the PI3 kinase VPS34 to generate PI(3)P as a recruitment signal for WIPI proteins, the mammalian orthologs of yeast Atg18 (Proikas-Cezanne *et al*, 2004). Since wortmannin, a potent inhibitor of VPS34, prevents colocalization of WIPI1 but not LC3 with *S.* Typhimurium (Kageyama *et al*, 2011), we tested whether TBK1 was similarly required for the recruitment of WIPI proteins to bacteria. We found that GFP-tagged WIPI1 and WIPI2B but not WIPI3 and WIPI4 accumulated on *S.* Typhimurium,

as did endogenous WIPI2 (Figs 2B and C, and EV2A and B). The recruitment of WIPI1 and WIPI2B was sensitive to wortmannin treatment and abrogated by mutations in their PI(3)P-binding sites (GFP:WIPI1$_{FTTG}$ and GFP:WIPI2B$_{FTTG}$). Importantly, accumulation of WIPI1 and WIPI2B also required expression of wild-type TBK1 and was not supported by either catalytically inactive $TBK1_{K38M}$ or $TBK1_{\Delta C}$ deficient in binding adaptor proteins (Fig 2B). In contrast, recruitment of DFCP1 did not require TBK1, although it was also sensitive to wortmannin treatment and mutational inactivation of its PI(3)P-binding site (DFCP1$_{FYVE*}$) (Fig 2D–F). WIPI proteins also bind PI(3,5)P$_2$ (Baskaran *et al*, 2012). However, the accumulation of PI(3,5)P$_2$ on *S.* Typhimurium was independent of TBK1, as revealed by the normal recruitment of GFP:ML1N*2, a PI(3,5)P$_2$-specific probe (Li *et al*, 2013b) (Fig 2G and H). We therefore conclude that TBK1 and VPS34 independently control the recruitment of WIPI1 and WIPI2 to *S.* Typhimurium and that TBK1 functionality requires catalytic activity as well as its C-terminal adaptor-binding coiled-coil domain.

To further investigate the mechanism of how TBK1 recruits WIPI1 and WIPI2 to cytosol-invading bacteria, we depleted cells of optineurin, the only known TBK1 substrate in anti-bacterial autophagy (Wild *et al*, 2011). Cells lacking optineurin recruited WIPI1 and WIPI2B normally to *S.* Typhimurium, suggesting that phosphorylation of a substrate other than optineurin is essential for WIPI1/2 recruitment in anti-bacterial autophagy (Fig EV2C). We also tested the interdependence of WIPI1 and WIPI2B recruitment and found that neither protein was required for the recruitment of the other (Fig EV2D).

We next investigated whether WIPIs are essential to protect cells against bacterial proliferation. Cells lacking WIPI2 failed to restrict proliferation of *S.* Typhimurium, confirming a recent finding (Dooley *et al*, 2014), while the presence of WIPI1 was not required (Fig 3). We therefore conclude that the recruitment of WIPI2 to cytosol-invading bacteria is likely an essential function of TBK1 in cell-autonomous defense.

## NDP52-mediated recruitment of TBK1 to *S.* Typhimurium suffices to restrict bacterial proliferation

Since the C-terminal domain of TBK1 is required to restrict bacterial proliferation and mediates adaptor binding (Fig EV1A) we thought to repair $TBK1_{\Delta C}$ by fusing it directly to individual adaptor proteins. This strategy enables the evaluation of individual adaptors in the TBK1-mediated restriction of *S.* Typhimurium and structure–function analyses without interference from potentially redundant adaptor function. As cytosol-exposed *S.* Typhimurium recruit Nap1 but not Tank (Thurston *et al*, 2009), we compared these two adaptors by fusing them to $TBK1_{\Delta C}$. Consistent with their differential recruitment to cytosolic *Salmonella*, $TBK1_{\Delta C}$:Tank did not restrict *Salmonella* proliferation in $Tbk1^{-/-}$ MEFs; in contrast, $TBK1_{\Delta C}$:

**Figure 2.  TBK1 kinase activity and C-terminal adaptor-binding domain are required to recruit WIPI1 and WIPI2 to *Salmonella enterica* serovar Typhimurium.**

A–H    Analysis of $Tbk1^{-/-}$ MEFs stably expressing the indicated TBK1 alleles and infected with *S.* Typhimurium for 1 h. Percentage of *S.* Typhimurium coated with GFP: LC3B (A), the indicated GFP:WIPI alleles (B), GFP:DFCP1 (FYVE* denotes a PI(3)P-binding mutant of DFCP1) (D, E), or GFP:ML1N*2, a probe for PI(3,5)P$_2$ (G). Where indicated, wortmannin (Wort) was added at 100 nM. Confocal micrographs of MEFs expressing the indicated GFP fusion proteins or immunolabeled for WIPI2 and infected with mCherry-expressing *S.* Typhimurium (C, F and H). Mean and SEM of at least three independent experiments with duplicate coverslips. > 200 bacteria counted per coverslip. *$P$ < 0.05, **$P$ < 0.01 one-way ANOVA with Dunnett's multiple comparisons test. Scale bar, 10 μm.

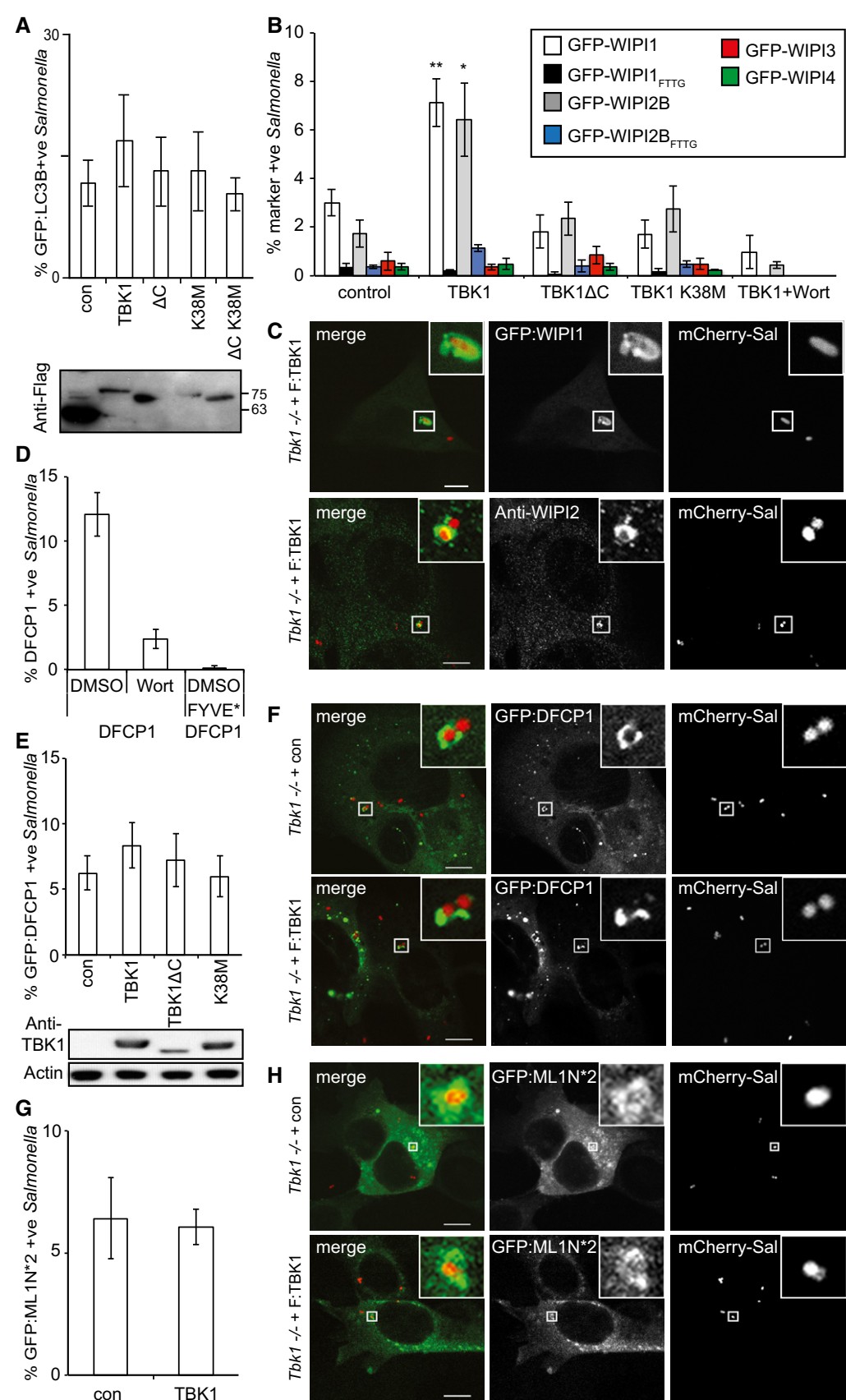

**Figure 2.**

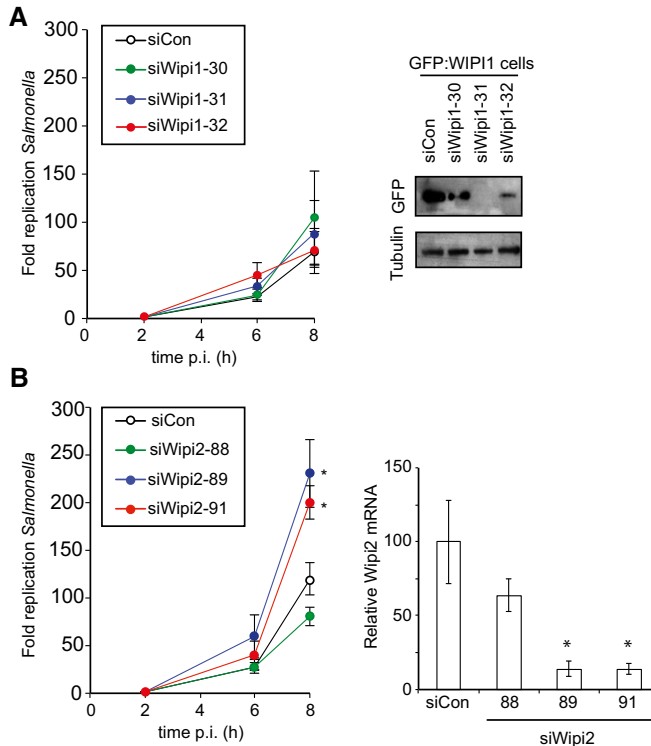

**Figure 3.   WIPI2 restricts *Salmonella* proliferation.**

A, B   Kinetics of *Salmonella enterica* serovar Typhimurium replication in MEFs depleted of WIPI1 (A) or WIPI2 (B). Intracellular bacteria were enumerated by their ability to form colonies on agar plates. Western blot for GFP:WIPI1 and quantitative RT–PCR for WIPI2 upon the indicated siRNA treatments. Mean and SEM. $N = 6$ (A), $n = 3$ (B). *$P < 0.05$, Student's *t*-test.

Nap1 reduced bacterial replication as efficiently as full-length TBK1 (Figs 4A and EV3A). Complementation of *Tbk1*$^{-/-}$ MEFs with TBK1$_{\Delta C}$:Nap1 but not with TBK1$_{\Delta C}$:Tank also restored localization of GFP-WIPI1 to *Salmonella* (Fig 4B).

How does Nap1 contribute to TBK1 function? Considering the overall structural similarity between Nap1 and Tank (Fig 4C) (Ryzhakov & Randow, 2007), efficient complementation with TBK1$_{\Delta C}$:Nap1 but not TBK1$_{\Delta C}$:Tank suggested a role for the N-terminal coiled-coil region of Nap1. Indeed, TBK1$_{\Delta C}$:Nap1$_{\Delta N85}$, in contrast to TBK1$_{\Delta C}$:Nap1, did not prevent hyper-proliferation of *S.* Typhimurium in *Tbk1*$^{-/-}$ MEFs (Figs 4A and EV3A). Complementation with TBK1$_{\Delta C}$:Nap1$_{N85}$ also restricted bacterial proliferation; the N-terminal 85 residues of Nap1 are therefore required and sufficient to provide functionality to TBK1$_{\Delta C}$.

Nap1$_{N85}$ forms a coiled-coil that contributes to the dimerization of Nap1 and binds the autophagy cargo receptor NDP52 (Thurston *et al*, 2009). We therefore speculated that recruitment of TBK1 to *Salmonella* via NDP52 antagonizes bacterial replication. To test this hypothesis, we fused TBK1$_{\Delta C}$ directly to NDP52 and found that it restricted bacterial proliferation in *Tbk1*$^{-/-}$ MEFs as efficiently as full-length TBK1 (Figs 4D and EV3B). However, although endogenous Nap1 cannot bind TBK1$_{\Delta C}$ directly (Fig EV1B), Nap1 may be recruited to TBK1$_{\Delta C}$:NDP52 via its binding site in the NDP52 SKICH domain. To test whether such indirect recruitment of Nap1

was required for the complementation of *Tbk1*$^{-/-}$ MEFs with TBK1$_{\Delta C}$:NDP52, we examined TBK1$_{\Delta C}$:NDP52$_{SKICH}$, which was inactive, and TBK1$_{\Delta C}$:NDP52$_{\Delta SKICH}$, which retained activity (Figs 4D and EV3B). We therefore conclude that in anti-bacterial autophagy, the interaction of TBK1 with adaptor proteins can be replaced entirely by fusing TBK1 directly to NDP52.

### Recruitment of TBK1 to *S.* Typhimurium via either galectin-8 or ubiquitin suffices to restrict bacterial proliferation

We speculated that the function of NDP52 in TBK1$_{\Delta C}$:NDP52 might be provided by its ability to sense cytosol-invading bacteria via autophagy-inducing "eat-me" signals, that is the bacterial ubiquitin coat and/or galectin-8 on damaged bacteria-containing vacuoles. Binding of NDP52 to galectin-8 is understood in structural detail; it is mediated by a hook-like structure formed by residues 371–381 and is abrogated in NDP52$_{L374A}$ (Kim *et al*, 2013; Li *et al*, 2013a). NDP52 binds ubiquitin via its C-terminal zinc finger and structural information on the interaction has been recently published (Xie *et al*, 2015). We also determined the solution structure of the C-terminal ubiquitin-binding zinc finger of NDP52 by NMR spectroscopy, which confirmed the existence of an UBZ-like fold consisting of an α-helix and a two-stranded β-sheet (Fig 5A) (Xie *et al*, 2015). The zinc ion is coordinated by residues His$_{440}$ and His$_{444}$ as well as residues Cys$_{422}$ and Cys$_{425}$, which are located in the helix and in the loop connecting the two β-strands, respectively. This fold is found in several ubiquitin-binding proteins with the NDP52 structure most similar to the C-terminal ubiquitin-binding Zn fingers of Nemo and optineurin (Fig 5B). Addition of mono-ubiquitin to the NDP52 zinc finger produced changes in the chemical shift primarily of residues in the helix (Fig 5C and Appendix Fig S1). Analysis of the chemical shift changes as a function of ubiquitin concentration revealed a dissociation constant of 60 μM (Appendix Fig S2). Perturbed ubiquitin residues upon binding to NDP52 are centered on the so-called Ile$_{44}$ patch in ubiquitin, a common binding site for UBDs (Fig 5C and Appendix Fig S1). These data suggest that the NDP52 zinc finger interacts with ubiquitin similar to other UBZ domains, that is via binding of the exposed face of the helix to the hydrophobic Ile$_{44}$ patch on ubiquitin. Consistent with such a binding mode and a crystallographic analysis of the NDP52 Zn finger bound to ubiquitin (Xie *et al*, 2015), residues on the exposed face of the helix are highly conserved among NDP52 orthologs (Fig 5D). To test our model, we mutated Asp$_{439}$, a conserved residue with a large ubiquitin-induced chemical shift change located on the exposed helix surface (Appendix Fig S3). NDP52$_{D439K}$ failed to bind ubiquitin (GST-4xUB) similar to NDP52D439R (Xie *et al*, 2015), while the interaction with GST-galectin-8 was maintained (Fig 5E). In contrast, NDP52$_{L374A}$ bound GST-4xUb but not GST-galectin-8. As expected, the double-mutant NDP52$_{L374A + D439K}$ did not bind to either ligand. We used the NDP52 mutants specifically deficient in binding to galectin-8 or ubiquitin to test (i) whether the recruitment of TBK1 to cytosol-invading *Salmonella* via bacteria-associated danger signals is essential to restrict bacterial growth and (ii) whether recruitment via both galectin-8 and ubiquitin is required. Bacterial proliferation in *Tbk1*$^{-/-}$ MEFs was potently restricted by both TBK1$_{\Delta C}$:NDP52$_{D439K}$ and TBK1$_{\Delta C}$:NDP52$_{L374A}$, which selectively bind galectin-8 or ubiquitin, respectively (Figs 5F and EV3B). In contrast, TBK1$_{\Delta C}$:NDP52$_{L374A + D439K}$, which binds neither galectin-8 nor ubiquitin,

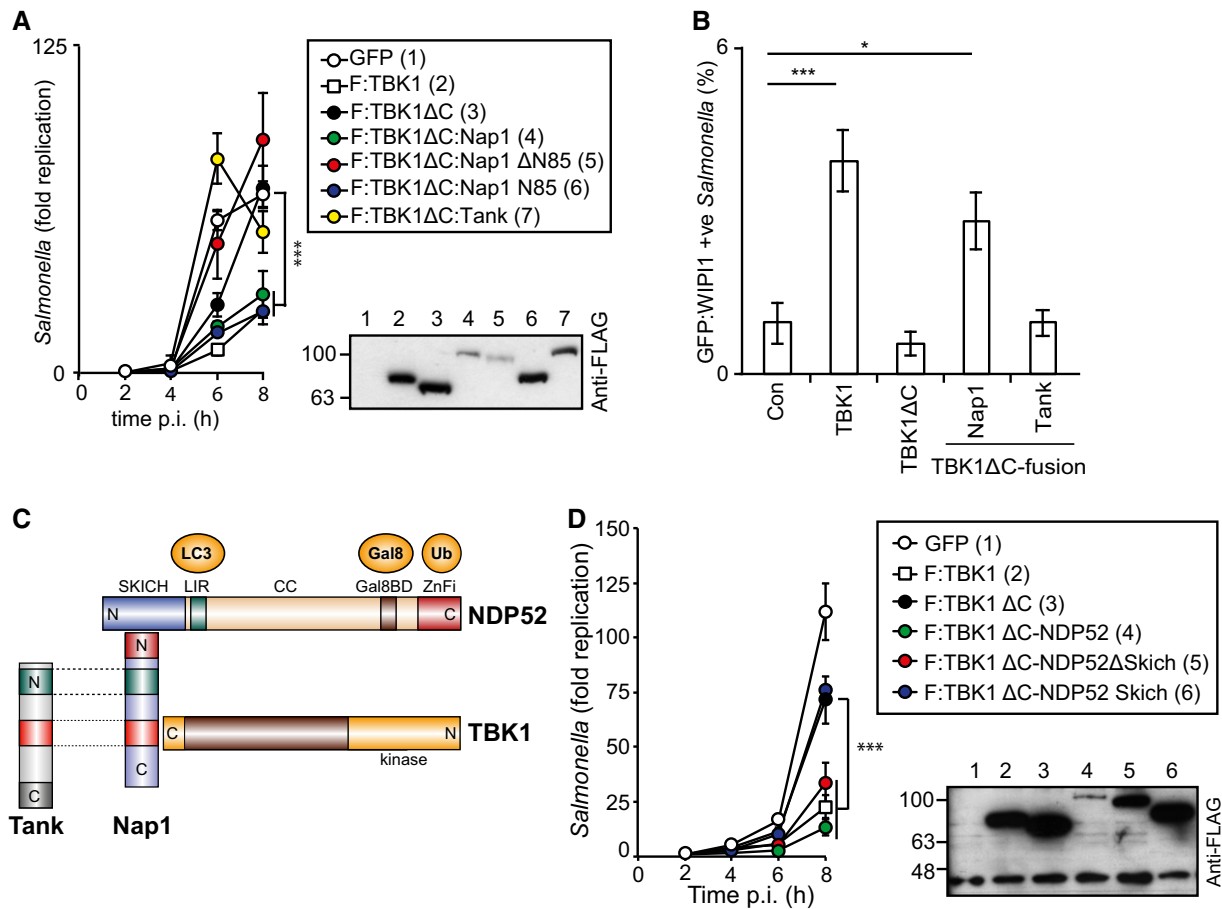

**Figure 4.  Recruitment of TBK1 to *Salmonella enterica* serovar Typhimurium via NAP1 or NDP52, but not TANK, restricts bacterial proliferation and recruits WIPI1.**

A–D   Analysis of $Tbk1^{-/-}$ MEFs complemented with the indicated Flag-tagged $TBK1_{\Delta C}$-adaptor fusion proteins. *S*. Typhimurium replication kinetics (A, D). Infected cells were lysed at the indicated time points post-inoculation (p.i.), and bacteria were enumerated by their ability to form colonies on agar plates. Mean and SD of triplicate MEF cultures and duplicate colony counts. Data are representative of at least two repeats. Statistical significance to $Tbk1^{-/-}$ MEFs expressing $TBK1_{\Delta C}$ is indicated. Western blot for Flag-tagged TBK1 variants in post-nuclear cell lysates. (B) Percentage of GFP:WIPI1-positive *S*. Typhimurium at 1 h p.i. in the indicated complemented MEFs. Mean and SEM of four independent experiments. > 200 bacteria counted per coverslip. *$P < 0.05$, ***$P < 0.001$, one-way ANOVA with Dunnett's multiple comparisons test. (C) TBK1–Nap1–NDP52 complex. N- and C-termini, domains and binding partners are indicated.

was inactive in this assay. We therefore conclude that the recruitment of TBK1 to cytosol-invading *Salmonella* via galectin-8 or the bacterial ubiquitin coat is essential to restrict bacterial proliferation and that either signal, that is galectin-8 or the ubiquitin coat, suffices in recruiting TBK1. To directly test this prediction for galectin-8, we complemented $Tbk1^{-/-}$ MEFs with $TBK1_{\Delta C}$ fused to galectin-8 ($TBK1_{\Delta C}$:Gal8). Indeed, $TBK1_{\Delta C}$:Gal8, similar to wild-type TBK1 but unlike $TBK1_{\Delta C}$, prevented hyper-proliferation of *S*. Typhimurium and enabled the recruitment of GFP:WIPI1 to bacteria (Figs 6 and EV4A).

### Recruitment of TBK1 to *S*. Typhimurium via K48- or K63-linked ubiquitin chains suffices to restrict bacterial proliferation

To test whether the bacterial ubiquitin coat also provides recruitment signals for TBK1 sufficient to protect cells against hyper-proliferation of *Salmonella*, we fused ubiquitin-binding domains to $TBK1_{\Delta C}$. The ubiquitin coat of *S*. Typhimurium contains

different linkage types; at minimum, M1- and K63-linked chains are present but their functional contribution to anti-bacterial autophagy remains unknown (van Wijk *et al*, 2012). NDP52 can bind M1-, K48-, and K63-linked ubiquitin chains (Wild *et al*, 2011) with similar affinities for di-ubiquitin molecules and tetra-ubiquitin chains that are M1-linked (Xie *et al*, 2015). To investigate whether NDP52 binds longer chains in a linkage-specific manner, we exposed NDP52 to an equimolar mixture of M1-, K48-, and K63-linked ubiquitin tetramers (Fig 7A). Control proteins of known linkage specificity (Trempe *et al*, 2005; Kulathu *et al*, 2009; Rahighi *et al*, 2009b) precipitated their ligands selectively; the UBAN domain of Nemo bound M1-linked chains, the NZF domain of TAB2 K63-linked chains, and the UBA domain of Mud1 K48-linked chains. In contrast, NDP52 bound equally well to all three ubiquitin chains, suggesting lack of linkage specificity for NDP52, at least among the tested linkages types.

To directly test the importance of individual ubiquitin-linkage types in recruiting TBK1 to the ubiquitin coat of cytosolic

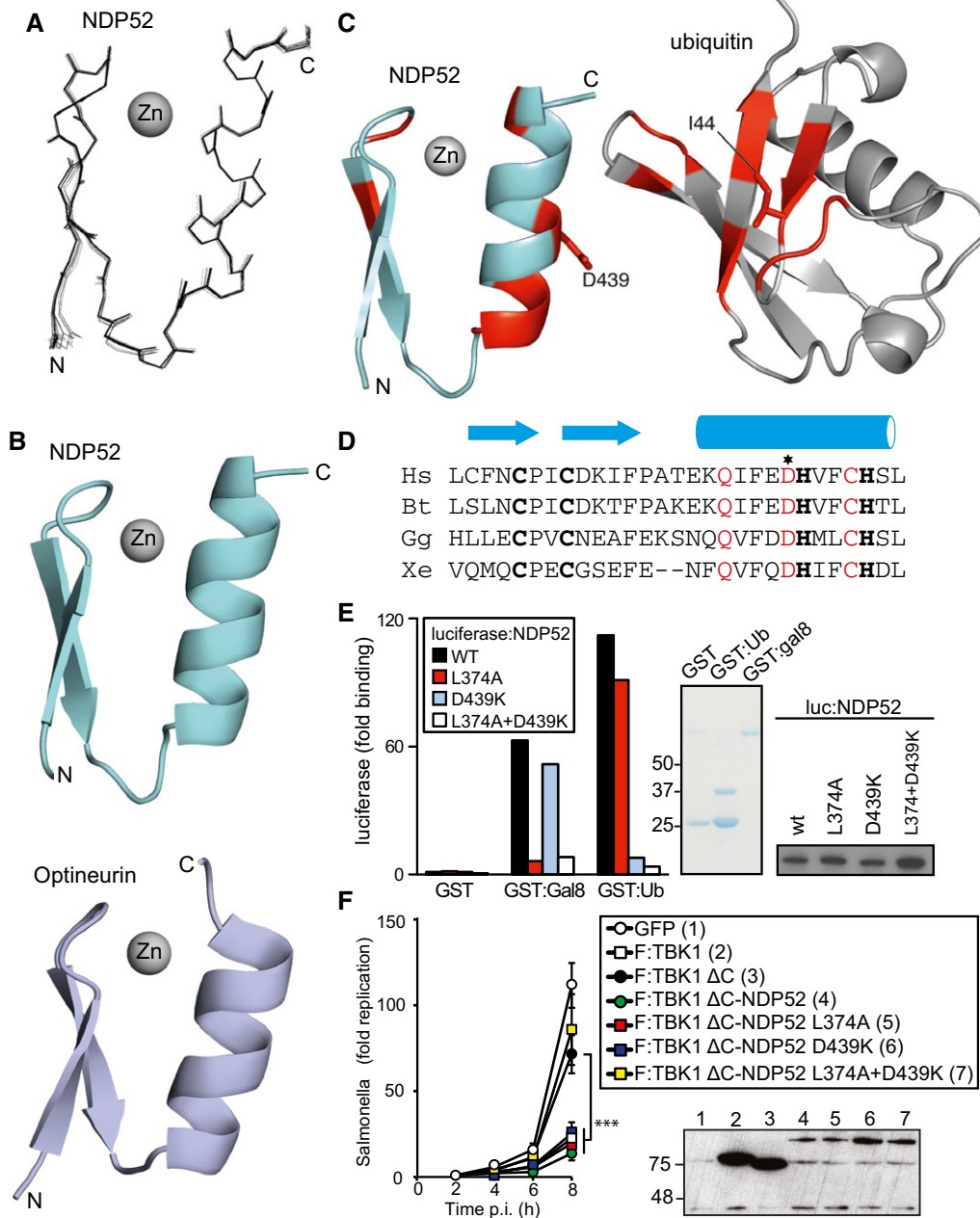

**Figure 5. Structure of the NDP52 zinc finger domain.**

A   Superposition of the 20 lowest energy solution structures of the NDP52 domain

B   Cartoon representation of the zinc finger domains of NDP52 and optineurin (pdb id: 5AAZ)

C   Cartoon representation of the NDP52 zinc finger and ubiquitin. In red are those residues whose signals alter upon binding.

D   Phylogenetic alignment of the NDP52 zinc finger. Zinc-coordinating residues in bold, conserved residues in the helix in red. The asterisk indicates D439, the residue mutated in subsequent experiments. Hs: *Homo sapiens*, Bt: *Bos taurus*, Gg: *Gallus gallus*, and Xe: *Xenopus laevis*.

E   Lumier binding assay. Binding of the indicated luciferase-tagged NDP52 variants to beads coated with GST, GST:galectin-8, or GST:ubiquitin. Coomassie stain of purified GST proteins. Western blot for luciferase-tagged NDP52 alleles in total cell lysates.

F   Kinetics of *Salmonella enterica* serovar Typhimurium replication in *Tbk1*$^{-/-}$ MEFs complemented with the indicated Flag-tagged TBK1$_{\Delta C}$-NDP52 fusion proteins. Infected cells were lysed at the indicated times post-inoculation (p.i.) and bacteria were enumerated by their ability to form colonies on agar plates. Mean and SD of triplicate MEF cultures and duplicate colony counts. Data are representative of at least two repeats. Statistical comparison with *Tbk1*$^{-/-}$ MEFs expressing TBK1$_{\Delta C}$ is indicated. ***$P < 0.001$, one-way ANOVA with Dunnett's multiple comparisons test. Western blot for Flag-tagged TBK1 variants in post-nuclear cell lysates.

*S.* Typhimurium, we fused TBK1$_{\Delta C}$ to ubiquitin-binding proteins of defined specificity. Complementing *Tbk1*$^{-/-}$ MEFs with TBK1$_{\Delta C}$: Nemo restricted the proliferation of *S.* Typhimurium (Figs 7B and EV5A). The UBAN domain of Nemo selectively binds M1-linked ubiquitin (Rahighi *et al*, 2009a), while full-length Nemo engages additional chain types due to combined contributions from its UBAN

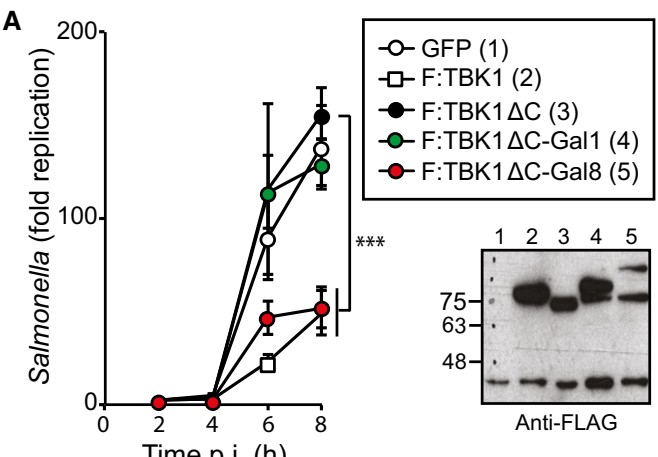

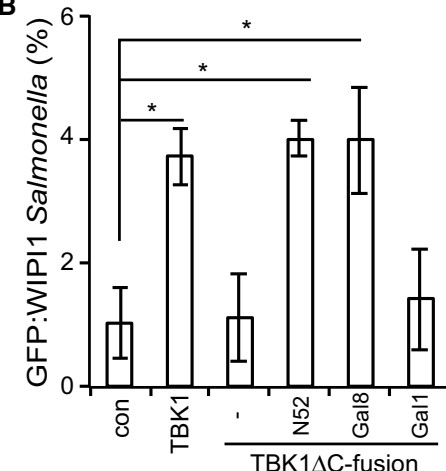

**Figure 6.  Recruitment of TBK1 to *Salmonella enterica* serovar Typhimurium via galectin-8 restricts bacterial proliferation and recruits WIPI1.**

A, B  Analysis of $Tbk1^{-/-}$ MEFs complemented with the indicated Flag-tagged TBK1$_{\Delta C}$-galectin fusion proteins. *S.* Typhimurium replication kinetics (A). Infected cells were lysed at the indicated times post-inoculation (p.i.), and bacteria were enumerated by their ability to form colonies on agar plates. Mean and SD of triplicate MEF cultures and duplicate colony counts. Data are representative of at least two repeats. Statistical differences to $Tbk1^{-/-}$ MEFs expressing TBK1$_{\Delta C}$ are shown. ***$P < 0.001$, one-way ANOVA with Dunnett's multiple comparisons test. Western blot for Flag-tagged TBK1 variants in post-nuclear cell lysates. (B) Percentage of GFP:WIPI1-positive *S.* Typhimurium at 1 h p.i. in the indicated complemented MEFs. Mean and SEM of three independent experiments. > 200 bacteria counted per coverslip. *$P < 0.05$, one-way ANOVA with Dunnett's multiple comparisons test.

domain and C-terminal Zn finger (Hadian *et al*, 2011). Consistent with a role for ubiquitin, and possibly linear chains, in recruiting TBK1 to cytosol-invading bacteria, TBK1$_{\Delta C}$:Nemo$_{D304N}$ carrying a point mutation in the UBAN domain as well as TBK1$_{\Delta C}$:Nemo$_{\Delta ZnF}$ failed to antagonize bacterial growth. Because of its homology to Nemo and its ability to bind TBK1 directly (Morton *et al*, 2008), we complemented $Tbk1^{-/-}$ MEFs with TBK1$_{\Delta C}$:optineurin, which prevented bacterial hyper-proliferation (Figs 7C and EV5B). Similar to TBK1$_{\Delta C}$:Nemo$_{D304N}$, TBK1$_{\Delta C}$:optineurin$_{D474N}$ was inactive due to loss of ubiquitin binding via its UBAN domain, while TBK1$_{\Delta C}$:optineurin$_{\Delta LIR}$ remained active. Recruitment of TBK1 via LC3/GABARAP, in contrast to ubiquitin-mediated recruitment is therefore insufficient to productively enroll TBK1 against cytosol-invading *S.* Typhimurium, likely because TBK1 function is required for an essential step in autophagy upstream of LC3/GABARAP conjugation to the phagophore, such as the recruitment of WIPI2. To assess whether K48- or K63-linked ubiquitin chains would be sufficient in recruiting TBK1 to cytosol-exposed *S.* Typhimurium, we deployed the ubiquitin-binding domains of Mud1 or TAB2, respectively. Complementation of $Tbk1^{-/-}$ MEFs with either TBK1$_{\Delta C}$:Mud1$_{UBA}$ or TBK1$_{\Delta C}$:TAB2$_{NZF}$ prevented hyper-proliferation of *S.* Typhimurium (Figs 7D and EV5C). Taken together, recruitment of TBK1 to cytosol-invading *S.* Typhimurium via K48- or K63-linked ubiquitin chains is sufficient to control TBK1 activity in anti-bacterial autophagy while conscription via LC3/GABARAP is not.

## Discussion

Our data establish that the recruitment of TBK1 to *S.* Typhimurium is essential to restrict bacterial proliferation by autophagy and that a variety of bacteria-associated "eat-me" signals, including

galectin-8 and K48- and K63-linked ubiquitin chains, can mediate TBK1 recruitment. Promiscuity in recruiting TBK1 may be of evolutionary advantage to the host by providing a backup mechanism against potential bacterial interference. We find that the localization of TBK1 activity to *S.* Typhimurium is required to control an upstream step in anti-bacterial autophagy, namely the recruitment of the ATG18 orthologue WIPI2 to cytosol-invading bacteria.

TBK1 is essential for cell-autonomous immunity against both viral and bacterial infections. In anti-viral immunity, TBK1 acts downstream of several important receptors—Toll-like receptors, RIG-like receptors, and cGAS—to phosphorylate IRF3 (Wu & Chen, 2014). IRF3 controls expression of type I interferons and other anti-viral genes that establish the so-called anti-viral state of increased resistance against a broad spectrum of viruses. In cell-autonomous anti-bacterial immunity, TBK1 specifically protects the host cytosol independently of IRF3 (Radtke *et al*, 2007; Thurston *et al*, 2009). While originally thought to control vesicular integrity and thus access of bacteria to the cytosol (Radtke *et al*, 2007), the unchanged recruitment of galectins to *S.* Typhimurium in cells lacking TBK1 placed TBK1 firmly downstream of bacterial entry (Thurston *et al*, 2012), while the epistatic relationship between ATG5 and TBK1 revealed in this study demonstrates that TBK1 protects cells against cytosol-invading *S.* Typhimurium by controlling autophagy. Consistent with TBK1 controlling anti-bacterial autophagy is the enzyme's ability to phosphorylate functionally important sites in cargo receptors, namely the LC3-interacting region (LIR) of optineurin (Wild *et al*, 2011) and the ubiquitin-binding site of p62 (Pilli *et al*, 2012), thereby enhancing their affinity for LC3 and ubiquitin, respectively. While the spatial and temporal control of the tethering function of cargo receptors appears as an important contribution of TBK1 to anti-bacterial autophagy, we found that in $Tbk1^{-/-}$ MEFs, LC3 is

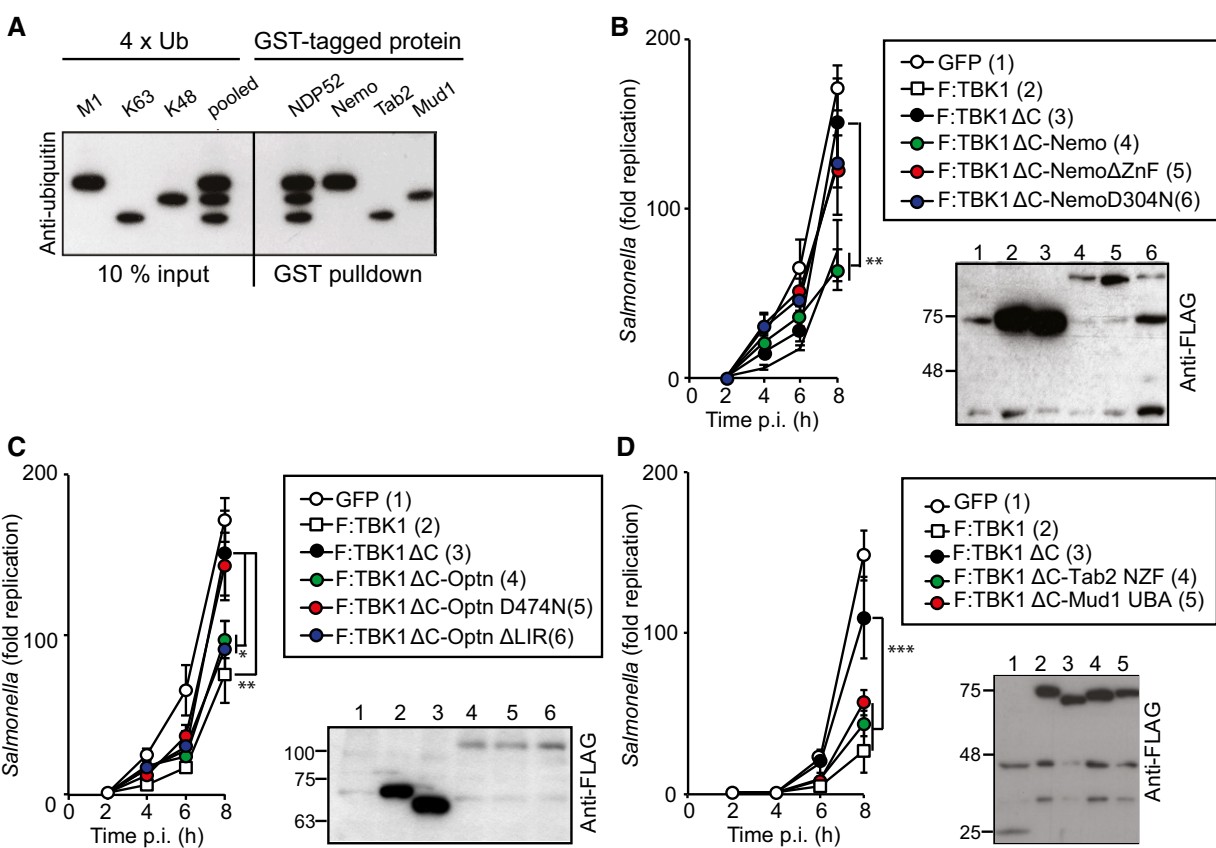

**Figure 7.** Recruitment of TBK1 to *Salmonella enterica* serovar Typhimurium via K48- or K63-ubiquitin chains restricts bacterial proliferation.

A     Binding of NDP52 to ubiquitin tetramers of different linkage types. Beads coated with the indicated GST-tagged proteins were incubated with pooled M1-, K63-, and K48-linked tetra-ubiquitin chains. Anti-ubiquitin Western blot of input and proteins bound to beads.

B–D   Kinetics of *S.* Typhimurium replication in *Tbk1*$^{-/-}$ MEFs complemented with the indicated Flag-tagged TBK1$_{\Delta C}$ fusion proteins. Infected cells were lysed at the indicated time points post-inoculation (p.i.) and bacteria were enumerated by their ability to form colonies on agar plates. Mean and SD of triplicate MEF cultures and duplicate colony counts. Data are representative of at least two repeats. Statistical differences to *Tbk1*$^{-/-}$ MEFs expressing TBK1$_{\Delta C}$ are shown. *$P < 0.5$, **$P < 0.01$, ***$P < 0.001$, one-way ANOVA with Dunnett's multiple comparisons test. Western blot for Flag-tagged TBK1 variants in post-nuclear cell lysates.

still recruited to *S.* Typhimurium. Apparently normal LC3 recruitment while failing to control the growth of cytosolic *S.* Typhimurium also occurs in MEFs deficient in FIP200 and ATG9, where it is caused by conjugation of LC3 to the damaged membrane of SCVs rather than to *de novo* phagophores attacking bacteria (Kageyama *et al*, 2011). Lack of TBK1, and failure to phosphorylate the LIR motif in optineurin, could result in a similarly subtle but nevertheless functionally important mislocalization of LC3.

However, irrespective of the precise membrane localization of LC3 in *Tbk1*$^{-/-}$ MEFs, the lack of WIPI2 recruitment, whose essential role in anti-bacterial autophagy we confirm (Dooley *et al*, 2014), provides a sufficient and more parsimonious explanation for the failure of autophagy to control bacterial proliferation. WIPI2 is recruited to bacteria via PI(3)P, which initially suggested insufficient PI(3)P in the bacterial vicinity may hinder anti-bacterial autophagy in *Tbk1*$^{-/-}$ MEFs. During autophagy PI(3)P is produced by the VPS34 lipid kinase complex and inhibition of VPS34 with wortmannin or knockout of ATG14L, a subunit of the VPS34 complex, abrogates anti-bacterial autophagy (Kageyama *et al*, 2011). However, a defect in PI(3)P production in *Tbk1*$^{-/-}$ MEFs can be ruled out since

DFCP1, a *bona fide* probe for the autophagic pool of PI(3)P, is recruited normally to *S.* Typhimurium. A defect in PI(3,5)P$_2$ levels (Baskaran *et al*, 2012), an alternative recruitment signal for WIPI2, can also be excluded since recruitment of GFP:ML1N*2, a PI(3,5)P$_2$-specific probe, did not depend on TBK1 either. We therefore conclude that an essential function of TBK1 in anti-bacterial autophagy is to control WIPI2 recruitment to cytosol-invading *Salmonella* independently of PIP-related signals. We propose that WIPI2 detects the coincidence of TBK1 and VPS34-generated signals on cytosol-invading bacteria in order to direct autophagy toward cognate cargo and away from cellular structures. The requirement for both TBK1 kinase activity and TBK1 localization to *S.* Typhimurium for the recruitment of WIPI2 furthermore suggests that the TBK1 substrates crucial for anti-bacterial autophagy are also located in the vicinity of the invading bacterium. Further experiments will be required to reveal their identity.

Anti-bacterial autophagy targets cytosol-invading bacteria in response to several distinct "eat-me" signals (Boyle & Randow, 2013). The earliest known "eat-me" signal comprises galectin-8, which detects damage to SCV membranes by binding

cytosol-exposed host glycans liberated by *S.* Typhimurium during its entry into the cytosol (Thurston *et al*, 2012). Galectin-8 is a selective ligand for the cargo receptor NDP52, which is physically linked to TBK1 via Nap1 and Sintbad, two homologous adaptor proteins (Ryzhakov & Randow, 2007; Thurston *et al*, 2009, 2012). The bacterial ubiquitin coat that develops subsequent to SCV damage comprises another "eat-me" signal (Perrin *et al*, 2004). Two E3 ligases, Parkin and LRSAM1, have been suggested to generate the bacterial ubiquitin coat, possibly in response to different bacterial species (Huett *et al*, 2012; Manzanillo *et al*, 2013). The ubiquitin "eat-me" signal is detected by several cargo receptors, of which optineurin and NDP52 provide independent physical links to TBK1. To overcome the redundancy in TBK1 recruitment regarding both "eat-me" signals and cargo receptors, we generated TBK1$_{\Delta C}$, a TBK1 allele unable to bind any of its known adaptors and thus unable to sense invading bacteria. TBK1$_{\Delta C}$ failed to complement $Tbk1^{-/-}$ MEFs, suggesting that adaptor binding is essential for TBK1 function in anti-bacterial autophagy. Importantly, TBK1 functionality was reconstituted by fusing TBK1$_{\Delta C}$ directly to NDP52 or optineurin but not to mutants deficient in binding "eat-me" signals, thus providing strong evidence that TBK1 recruitment to "eat-me" signals is essential for autophagy against cytosol-invading *S.* Typhimurium. Fusing TBK1$_{\Delta C}$ to NDP52 alleles selectively deficient in ubiquitin binding or directly to galectin-8 demonstrated that glycan-driven recruitment suffices to provide TBK1 functionality for anti-bacterial autophagy. In contrast, NDP52$_{C425A}$, a mutation that destabilizes the C-terminal zinc finger, has been reported not to support anti-bacterial autophagy because of a defect in phagosome maturation, an apparently TBK1-independent function of NDP52 (Verlhac *et al*, 2015).

Similarly, fusing TBK1$_{\Delta C}$ to optineurin or NDP52 deficient in galectin-8 binding revealed that ubiquitin-driven recruitment also suffices. Why would such apparent redundancy in recruiting TBK1 to invading bacteria be needed? Glycan- and ubiquitin-encoded signals can provide functionality to TBK1 for cytosolic defense against invading bacteria at different stages of infection. Recruitment of TBK1 via galectin-8 enables autophagy to catch invading bacteria as soon as they appear in the cytosol but would fail against bacteria that thwart the attack long enough for cytosol-exposed glycans to become degraded by cytosolic glycosidases or against those bacteria that proliferate and thereby outgrow the vacuolar membrane remnants. The ubiquitin coat, on the other hand, requires time to develop but could be maintained even on moderately proliferating bacteria for long enough that ultimately anti-bacterial autophagy may succeed. We therefore suggest that redundancy in TBK1 recruitment via the galectin-8 and ubiquitin route equips anti-bacterial autophagy with sufficient robustness to efficiently combat cytosol-invading bacteria throughout the invasion process.

Additional sturdiness to TBK1 recruitment is provided by the ability of NDP52 to bind M1-, K48-, and K63-linked ubiquitin chains similarly (Xie *et al*, 2015; this study), which enables anti-bacterial autophagy to sense a variety of ubiquitin-derived "eat-me" signals, for example those generated by different E3 ligases on bacterial and host-derived substrates. The ubiquitin coat on *S.* Typhimurium comprises, at minimum, M1- and K63-linked chains (van Wijk *et al*, 2012). By fusing linkage-specific ubiquitin-binding domains to TBK1$_{\Delta C,}$ we demonstrate that the ubiquitin coat on *S.* Typhimurium also contains K48-linked

chains and that recruitment of TBK1 via K48- and K63-linked chains suffices for anti-bacterial autophagy. The inability of the UBAN domain in NemoΔZnF to recruit TBK1 implies that M1-linked ubiquitin chains may not be a sufficient signal to provide functionality to TBK1. However, we refrain from explicitly ruling out that M1-linked chains could provide such a signal since the combined effect of zinc finger and adjacent UBAN domain on Nemo's affinity and specificity for M1 and other linkage types is insufficiently understood. Nevertheless, not all signals found on cytosol-invading *S.* Typhimurium are sufficient to provide functionality to TBK1$_{\Delta C}$, as demonstrated by the failure of NDP52 and optineurin to recruit TBK1 functionality to bacteria-associated LC3 via their LIR motifs.

Taken together, in this study, we demonstrated that recruitment of TBK1 to cytosol-invading bacteria is essential for anti-bacterial autophagy and that multiple recruitment signals provide TBK1 functionality for anti-bacterial autophagy throughout distinct stages of bacterial invasion. We suggest that such robustness in providing TBK1 functionality to anti-bacterial autophagy protects the pathway from potential bacterial antagonism and is of evolutionary advantage to the host.

## Materials and Methods

### Antibodies

Antibodies were from Enzo Life Science (ubiquitin FK2), Sigma (Flag M2), Abcam (WIPI2, ab101985; TBK1, ab40676), Dabco (HRP-conjugated reagents), and Invitrogen (Alexa-conjugated anti-mouse and anti-goat antisera).

### Bacteria

*Salmonella enterica* serovar Typhimurium (strain 12023) was grown overnight in LB and subcultured (1:33) in fresh LB for 3.5 h prior to infection. MEF cells in 24-well plates were infected with 5 µl of such cultures for 7 min. Following two washes with warm PBS and a 2-h incubation with 100 µg/ml gentamycin, cells were cultured in 20 µg/ml gentamycin. Where indicated, wortmannin (Sigma; 100 nM) or DMSO vehicle control was added 15 min prior to infection and maintained during the experiment. To enumerate intracellular bacteria, cells from triplicate wells were lysed in 1 ml cold PBS containing 0.1% Triton X-100. Serial dilutions were plated in duplicate on LB agar.

### Cell culture

$Tbk1^{-/-}$ MEFs (gift from Kate Fitzgerald, University of Massachusetts), $Atg5^{-/-}$ MEFs (gift from Noboru Mizushima, University of Tokyo), and 293ET cells (gift from Brian Seed, Harvard University), confirmed as mycoplasma free, were grown in IMDM supplemented with 10% FCS at 37°C in 5% $CO_2$. MEFs expressing GFP-tagged LC3B were obtained by limiting dilution of retrovirally transduced cells. MEFs expressing GFP-tagged WIPI proteins, DFCP1, or ML1N*2 were transduced with retrovirus and selected in blasticidin at 5 µg/ml. To complement $Tbk1^{-/-}$ MEFs, cells were transduced with retrovirus and selected in puromycin at 1.5 µg/ml.

    

## LUMIER assays

Binding assays with pairs of putative interactors, one fused to luciferase and the other fused to GST or Flag, were performed in Lumier lysis buffer (150 mM NaCl, 0.1% Triton X-100, 20 mM Tris–Cl (pH 7.4), 5% glycerol, 5 mM EDTA and protease inhibitors). GST fusion proteins were immobilized on beads before incubation with the luciferase-tagged binding partner for 2 h. For Flag-based assays, both proteins were expressed in 293ET cells and immobilized using Flag-agarose. After washing in lysis buffer, proteins were eluted with glutathione or Flag peptide in Renilla lysis buffer (Promega). Relative luciferase activity represents the ratio of activity eluted from beads and present in lysates.

## Ubiquitin-binding assay

Ubiquitin chains and assay conditions have been described previously (Komander *et al*, 2009).

## Western blot

Post-nuclear supernatants from $1 \times 10^6$ MEF cells or 293ET cells expressing the indicated fusion proteins were separated on a 12% denaturing Bis-Tris gels. Proteins were transferred to PVDF membrane and incubated with relevant primary and secondary antibodies. Visualization following immunoblotting was performed using ECL detection reagents (Amersham Bioscience).

## Microscopy

MEFs were grown on poly-L-lysine pre-treated glass cover slips prior to infection. Following infections, cells were washed twice with warm PBS and fixed in 4% paraformaldehyde in PBS for 20 min. Cells were washed twice in PBS and then quenched with PBS pH 7.4 containing 1 M glycine and 0.1% Triton X-100 for 30 min prior to blocking for 30 min in PBTB (PBS, 0.1% Triton X-100, 2% BSA). Where required, cover slips were incubated with primary antibody for 2 h, followed by secondary antibodies and DAPI (4′,6-diamidino-2-phenylindole) for 1 h in PBTB before being mounted (Vector Laboratories). Confocal images were taken with a 100× 1.4 objective on a Zeiss 710 microscope.

## Plasmids

M5P or closely related plasmids were used to express proteins in mammalian cells. For TBK1$_{\Delta C}$ fusions (N685), genes encoding full-length or deletion constructs of murine Nemo, Nap1, Tab2, or human NDP52, galectin-1, galectin-8, optineurin or Tank were amplified by PCR and ligated BspHI to NotI, in frame with TBK1$_{\Delta C}$. Mutations were generated by PCR and verified by sequencing. Open reading frames for LC3B, DFCP1, WIPI1, WIPI2B, WIPI3, and WIPI4 were amplified from human cDNA. PI(3)P-binding mutants in WIPI (FRRG-FTTG) were introduced by mutational PCR, that is WIPI1$_{RR226/227TT}$ and WIPI2$_{RR224/225TT}$. Plasmids encoding GFP-DFCP1 double FYVE mutant (FYVE*) and GFP-ML1N*2 have been described (Ridley *et al*, 2001; Li *et al*, 2013b).

| Construct name | Further construct information | References |
|---|---|---|
| TBK1 | Full-length TBK1 | |
| TBK1ΔC | TBK1 N685, lacks the C-terminal adaptor-binding region | |
| TBK1 K38M | TBK1 catalytic mutant | Pomerantz and Baltimore (1999) |
| TBK1ΔC K38M | TBK1 N685 K38M | |
| TBK1ΔC:Nap1 | TBK1 N685 fused to full-length Nap1 | |
| TBK1ΔC:Nap1 ΔN85 | TBK1 N685 fused to Nap1 ΔN85, Nap1 lacks the CC for binding NDP52 | Ryzhakov and Randow (2007) |
| TBK1ΔC:Nap1 N85 | TBK1 N685 fused to N85 of Nap1 | Ryzhakov and Randow (2007) |
| TBK1ΔC:Tank | TBK1 N685 fused to full-length Tank | |
| TBK1ΔC:NDP52 | TBK1 N685 fused to full-length NDP52 | |
| TBK1ΔC:NDP52 ΔSkich | TBK1 N685 fused to NDP52 ΔN127, NDP52 lacks the Skich domain | Gurung (2003) |
| TBK1ΔC:NDP52 Skich | TBK1 N685 fused to N127 of NDP52 (the Skich domain) | Gurung (2003) |
| TBK1ΔC:NDP52 L374A | TBK1 N685 fused to NDP52 L374A, a galectin-8-binding mutant of NDP52 | Thurston *et al* (2012), Li *et al* (2013a) |
| TBK1ΔC:NDP52 D439K | TBK1 N685 fused to NDP52 D439K, a ubiquitin-binding mutant of NDP52 | This study |
| TBK1ΔC:NDP52 L374A+D439K | TBK1 N685 fused to NDP52 L374A + D439K, a mutant of NDP52 lacking ubiquitin binding and galectin-8 binding | This study |
| TBK1ΔC:Gal1 | TBK1 N685 fused to full-length galectin-1 | |
| TBK1ΔC:Gal8 | TBK1 N685 fused to full-length galectin-8 | |
| TBK1ΔC:Nemo | TBK1 N685 fused to full-length Nemo | |
| TBK1ΔC:NemoΔZnF | TBK1 N685 fused to Nemo ΔC389, lacking the C-terminal ZnF | Bloor *et al* (2008) |
| TBK1ΔC:NemoD304N | TBK1 N685 fused to Nemo D304N, a ubiquitin-binding mutant of Nemo | Bloor *et al* (2008) |
| TBK1ΔC:Optn | TBK1 N685 fused to full-length optineurin | |
| TBK1ΔC:Optn D474N | TBK1 N685 fused to Optn D474N, a ubiquitin-binding mutant | Zhu *et al* (2007) |
| TBK1ΔC:Optn ΔLIR | TBK1 N685 fused to Optn F178S+ V179S+I181S, an LC3 interacting mutant | Wild *et al* (2011) |
| TBK1ΔC:Tab2 NZF | TBK1 N685 fused to Tab2 AA663–693 | Kulathu *et al* (2009) |
| TBK1ΔC:Mud1 UBA | TBK1 N685 fused to Mud1 UBA AA293–332 | Trempe *et al* (2005) |

## RNA interference

About $5 \times 10^4$ MEFs were reverse-transfected with 6 pmol of siRNA (Optn siRNA, Invitrogen) or 40 pmol siRNA against WIPI1 (Oligo

IDs MSS225230, MSS225231, and MSS225232, Invitrogen) or WIPI2 (Oligo IDs MSS232888, MSS232890, and MSS232891, Invitrogen), using Lipofectamine RNAiMAX (Invitrogen). Control siRNAs were purchased from Invitrogen. siRNA transfection was repeated after 2 days, and experiments were performed 4 days after the first siRNA treatment.

siOPTN #29 5′-GAAGCUAAAUAAUCAAGCU
siOPTN #30 5′-GCCUCGCAGUAUUCCGAUU

## Quantitative PCR

RNA was extracted (Qiagen) from siRNA-treated MEF cells and converted into cDNA (SuperScript III reverse transcriptase kit, Invitrogen) according to the manufacturer's protocol. SYBR Green qPCR kit (Applied Biosystems) was used to quantify gene expression using the following primer pairs:

muWipi2 5′-ATGAACCTGGCGAGCCAGAGC-3′
and 5′-GCTGGAGAACAATCTCTCTAC-3′
muRsp9 5′-CTGGACGAGGGCAAGATGAAGC-3′
and 5′-TGACGTTGGCGGATGAGCACA-3′

Data were normalized to Rsp9 levels in each sample after relative cDNA levels were calculated from a standard curve.

## NMR

NDP52 zinc finger domain samples for NMR spectroscopy experiments were typically at 1.0 mM in 90% $H_2O$ and 10% $D_2O$ in PBS containing 10 mM 2-mercaptoethanol. All spectra were acquired with either a Bruker Advance 700 or a DRX600 spectrometer at 20°C and referenced relative to external sodium 2,2-dimethyl-2-silapentane-5-sulfonate (DSS) for proton and carbon signals, or liquid ammonium for nitrogen. Assignments were obtained using standard NMR methods using $^{13}C/^{15}N$-labeled and $^{15}N$-labeled samples. Distance constraints were derived from 2D NOESY spectra recorded on 1.5 mM samples with a mixing time of 100 ms. The three-dimensional structure of the domain was calculated using the standard torsion angle dynamics-simulated annealing protocol in the program CNS 1.2. Structures were accepted where no distance violation was > 0.25 Å and no dihedral angle violations > 5°. To map the binding interface between the NDP52 zinc finger and ubiquitin, a series of $^1H$-$^{15}N$ HSQC spectra of $^{15}N$-labeled zinc finger domain in the presence of increasing molar ratios of unlabeled ubiquitin were recorded; a reciprocal titration was carried out with the unlabeled zinc finger domain added to $^{15}N$-labeled ubiquitin. Dissociation constants were obtained by fitting the concentration dependence of the normalized chemical shift changes to a single site-binding model.

## Accession numbers

Coordinates have been deposited under PDB code 5AAQ and 5AAZ.

## Statistical analysis

Student's *t*-test or one-way ANOVA with Dunnett's multiple comparisons test were used as indicated. *$P < 0.05$, **$P < 0.01$ and ***$P < 0.001$.

Expanded View for this article is available online.

## Acknowledgements

We thank Medical Research Council Technology for providing MRT68843. Work in the authors' laboratories was supported by the Medical Research Council (MC_U105170648, F.R., MC_U105192732, D.K.), the Wellcome Trust (WT104752MA, F.R.), the Leverhulme Trust (T.L.M.T.), the European Research Council (309756, D.K.), and the Lister Institute for Preventive Medicine (D.K.). We thank Drs. Haoxing Xu (University of Michigan, USA) for GFP:ML1N*2, Nicholas Ktistakis (Babraham Institute, Cambridge, UK) for GFP:DFCP1 double FYVE mutant, Kate Fitzgerald (Univsersity of Massachusetts Medical School) for $Tbk1^{-/-}$ MEFs, and Noboru Mizushima (University of Tokyo, Japan) for $Atg5^{-/-}$ MEFs.

## Author contributions

TLMT, KBB, BJR, MK, SB, AK, JN, AF, and SAM planned, performed, and analyzed the experiments; MA and MB performed and analyzed NMR experiments; DK provided reagents and advice; and TLMT and FR designed the overall research and wrote the manuscript.

## Conflict of interest

The authors declare that they have no conflict of interest.

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
