## [Review Process File · The EMBO Journal]

Manuscript EMBO-2016-94491

Recruitment of TBK1 to cytosol-invading Salmonella induces WIPI2-dependent antibacterial autophagy

Teresa LM Thurston, Keith Boyle, Mark Allen, Ben Ravenhill, Maryia Karpiyevich, Stuart Bloor, Annie Kaul, Jessica Noad, Agnes Foeglein, Sophie A Matthews, David Komander-Member, Mark Bycroft and Felix Randow

Corresponding author: Felix Randow, MRC Laboratory of Molecular Biology

Review timeline:

Submission date:	05 August 2015
Editorial Decision:	24 September 2015
Revision received:	07 April 2016
Editorial Decision:	13 May 2016
Revision received:	23 May 2016
Accepted:	24 May 2016

Editor: Andrea Leibfried

Transaction Report:

Preliminary Decision

15 September 2015

Thank you again for submitting your manuscript to us. I have now three reports on your work, which I copy below.

You will see that the referees appreciate your work. However, they also think that the insight into how TBK1 recruits WIPI and how this affects the upstream autophagy regulators remains rather unclear, and I agree with this view. Adding such insight would be needed for further consideration here. Addressing the concerns might be feasible with additional experiments as outlined by the three referees, who all provide constructive reports.

However, doing so might also represent a significant amount of work with uncertain outcome. Therefore, before taking a decision, I think it would be most productive if you could provide me upfront with a point-by-point response to the raised criticisms in order to see how you would address them. You could envision a 6 months revision time for doing so. Please send me the point-by-point draft by reply email - I am looking forward to receiving it before making a formal editorial decision. Thank you very much.

REFEREE REPORTS

Referee 1#:

In this manuscript, Thurston, et al, investigate a mechanism by which the kinase TBK1 regulates autophagy. Binding of NDP52 to ubiquitin engaged TBK1, which was necessary for recruitment of autophagy regulating factors. The authors found that linking TBK1 directly to several different

autophagy-related factors delineated specificity for downstream signaling components.

Overall, this work focuses on an important area of innate immune function that needs further elucidation, specifically how cytosolic bacteria trigger ubiquitin-mediated autophagy. In general, the manuscript is well written and the experiments executed with appropriate controls. The authors use elegant molecular biology approaches to identify critical downstream signaling mediators, and are nicely filling in the details of autophagy initiation. However, there are several major points of concern, indicated in more detail below. Additionally, although the authors do provide evidence for a more detailed hierarchy in initiating autophagy, the mechanism by which TBK1 recruits the upstream autophagy component, WIPI1, is hinted at, but still not addressed. These points, as well as the specific issues below, decreased my enthusiasm for the study in its present form.

Specific points the authors should address:

1. The magnitude of the phenotypes they showcase throughout this particular study are fairly modest, perhaps 1.5-3 fold, e.g., Fig. 2A. In their original paper (Thurston, Nat Imm 2009), the differences were much more striking. It was not evident why the phenotype quantitatively changed. In light of the concerns about the statistical analysis in point #4 below, this point should be addressed.
2. The authors suggest several times in the paper that the way in which TBK1 controls bacterial proliferation is unknown. Data in Fig. 1 lead the authors to suggest that TBK1 kinase activity may regulate VPS34 function, leading to the observed differences in WIPI1 recruitment, which would be an exciting finding to address their rationale, but this is not tested.
3. In Fig. 5B, the data show that only 4% or so associate with GFP-WIPI1, which the authors propose as the TBK1-dependent determining factor in promoting anti-bacterial autophagy (as opposed to non-productive LC3 association with SCV). This number seems quite low for a mechanism proposed to restrict infection.
4. Statistical concerns: for example, in Fig. 6C - the authors again emphasize statistically significant differences, e.g., between 100 and 150 fold replication, which seems quite modest. In this figure and others, the figure legend indicates the data are "representative" of at least 2 repetitions of the experiment. From what I can understand from the figure legends, the statistics are done on triplicate cell samples and duplicate colony counts within the same experiment. If this is correct, this is not the appropriate way to do statistical analysis. For more robust analysis, the statistical comparisons should be done with data across several experiments.

Referee #2:

The study submitted by Thurston et al. investigates the role of TBK1 in mediating bacterial autophagy. The same group previously showed that NDP52 is a specific receptor for anti-bacterial autophagy that recruits TBK1 through specific adaptors, and recognizes bacteria through galectin-8 binding (Thurston et al. 2009, Nature Immunology and Thurston et al. 2012, Nature). This study expands further on the role of TBK1 while also investigating the structural basis for NDP52 binding to ubiquitin. The study is of high quality and the authors' conclusions are generally well supported by their results. Of particular interest is the observations that ubiquitin- or Gal8-binding by NDP52 are sufficient to mediate bacterial autophagy. However, some aspects have been neglected, such as the mechanism of action of TBK1 kinase activity, beyond the requirement for adaptor-mediated localization. Thus, while this manuscript certainly contains interesting findings and reinforces our understanding of bacterial autophagy, I believe it should be published in a more specialized journal, unless more mechanistic data is provided. Here are some specific points for the authors to consider:

1. Perhaps the biggest outstanding question, which has not been explored, is the mechanism of action of TBK1. What is the phosphorylation step required for TBK1-mediated WIPI1 recruitment? TBK1 phosphorylates Optineurin (OPTN) at Ser177, in both bacterial autophagy (Wild et al. 2011), as well as mitophagy (Lazarou et al. 2015). Is OPTN, as well as its phosphorylation, required for WIPI1 recruitment? Can a phosphomimetic OPTN bypass TBK1 recruitment for this process? Alternatively, the authors speculate that TBK1 might target the VPS34 complex, which would then make PI(3)P and recruit WIPI1. Another possibility is the potential involvement of TBK1 in the

activation of the autophagy initiation complex (Atg1/ULK1 complex). These potential mechanisms can and should be tested experimentally.

2. The intent for testing exclusively LC3B recruitment is not clear (Figure 1D). NDP52 recruits LC3C; hence the recruitment of LC3C should also be tested in the context of TBK1 activity and function. Also, even though the recruitment of LC3B is normal in TBK1-null cells, an important factor to consider from the perspective of autophagy is to monitor the conversion of LC3 type I to type II forms. The authors also need to state the intent of this experiment more clearly: Rather than just saying 'visualize anti-bacterial autophagy attack', they should state whether they were expecting LC3B association to change and also provide the context for the use of LC3B as a marker with literature references.

3. The authors fail to acknowledge a study published earlier this year about the role of NDP52 during the early and late stages of bacterial autophagy (Verlhac et al. 2015, Cell Host and Microbe). In particular, it was observed that mutation of C425 in the ZnF domain of NDP52 fails to restrict bacterial proliferation in cells treated with NDP52 siRNA, suggesting that ubiquitin-binding is not dispensable for bacterial autophagy. This contradicts the current study, which shows that the mutation D439K alone does not affect the function of TBK1 Δ C-NDP52. One major difference is that endogenous NDP52 is still present in the latter case, which may complement the activity of the chimeric construct through dimerization. The authors should eliminate the possibility that the observed effects may be mediated by endogenous NDP52, and should try to reconcile their observations with this other study or explain otherwise.

4. In all experiments involving the fusion of TBK1 to different constructs of Nap1, NDP52, Gal8 or OPTN, control experiments should be performed with the cells transfected with only the respective inserts from Nap1, NDP52, Gal8 or OPTN, to eliminate the possibility that the observed effects find their origin in the overexpression of those inserts.

Minor points:

5. In the description of the results for figure 1A, the authors conclude that "adaptor binding controls kinase activity spatially and/or temporarily". While I understand what the authors meant, it may be interpreted as if the adaptor protein controlled the kinase activity itself, for which there is no evidence. The sentence needs to be rephrased for clarity. Also, it's 'temporally', not 'temporarily'.

6. Ubiquitination of *S. Typhimurium* was assessed after 1 and 4 hours (Figure 1C) while L3CB association was tested only 1 hour post-infection. Could a difference be observed 4 hour later? This needs to be justified especially since L3CB recruitment is a more 'downstream' process.

7. Figure 4 A, B and C: The authors should annotate the important residues on the figures, especially those that have been tested in assays. This is particularly essential for figure 4C in which chemical shift perturbations have been mapped and the interaction with ubiquitin is being shown. Besides the ubiquitin experts, no one can tell where the I44 patch is on the cartoon.

8. The manuscript needs to be checked for typos and labeling errors, such as "TKB1" on page 6. Also, there is no Figure 3B (p. 9). Finally, in Figure 6D, the labels should be "Tab2-ZnF" and "Mud1-UBA".

Referee #3:

The manuscript by Thurston et al. sheds new light on the mechanism of regulation of anti-bacterial autophagy, particularly focusing on an important player in this context, the kinase TBK1. Indeed, the authors show that recruitment of TBK1 to cytosolic Salmonella is mediated, with similar efficiency, by different "eat-me signals", and that such a re-localization of TBK1 is fundamental to the prevention of Salmonella proliferation.

Interestingly, TBK1 presumably causes an enrichment in Phosphatidylinositol 3-phosphate (PI3P) in the proximity of cytosolic bacteria, resulting into recruitment of WIPI1, an upstream regulator of the autophagy pathway. This finding opens an exciting scenario, suggesting an unexpected link between

TBK1 and the upstream regulation of autophagy, possibly functional to efficiently restrict bacteria proliferation.

Altogether, the manuscript is well written, and the rationale, the experiment set-up and the interpretation of the data are thoughtful and elegant. Also, all along the manuscript, the authors take advantage of numerous deletion and single-point mutants, as well as TBK1-fusion proteins, in order to elegantly prove the relevance of TBK1 localization to Salmonella and to dissect the function of different "eat-me-signals" in this context. On the one hand, such an approach represents a great effort and is impressive, although, on the one other hand, in some cases the conclusions drawn by the authors exclusively rely on mutants that may need further characterization before being considered trustful and being used for functional assays.

This aspect represents one of my main concerns regarding the manuscript and I report below my detailed suggestions to further improve this study.

Major comments

1. Figure S1. The coomassie stain of purified GST proteins and the Western Blot of luciferase-NDP52 alleles, used in this experiment, are necessary controls. For this reason, they should be shown in order to support the reliability of the experiment itself.

2. It is not clear to me why the authors did not test the binding capability of TBK1- Δ C construct for Sintbad and why did they keep this TBK1-binding partner out of their study.

3. Figure 1B. The authors show that the restriction of Salmonella proliferation is prevented by TBK1 inhibition (MRT68843) in Atg5^{-/-} reconstituted MEFs. First, a reference should be cited supporting the efficiency and specificity of MRT68843 inhibitor, which should also be mentioned in the Methods section. Also, in order to further confirm the result obtained by using the inhibitor, the authors should also assess Salmonella proliferation in Atg5^{-/-} and reconstituted MEFs upon TBK1 knock-down.

4. A truncated form of TBK1 (TBK1- Δ C) and its fusion proteins are extensively used all along the manuscript. As the authors properly tested, TBK1- Δ C is defective for binding to TBK1 adaptors (Figure S1). Nevertheless, the authors did not take into account the possibility that, messing up with TBK1 structure and localization, they could also accidentally affect TBK1 kinase activity, a possibility that is also clearly discussed in the literature (ref. in Helgason et al., 2013). On this line, the authors should check for the catalytic activity of TBK1- Δ C upon its over-expression in mammalian infected with Salmonella cells. Also, the phosphorylation levels of TBK1 substrates (such as p62, Optineurin, etc.) must be measured in TBK1^{-/-} MEFs reconstituted with WT, TBK1- Δ C and the kinase-dead TBK1.

Finally, given that TBK1 has been reported to dimerize, did the author test the possibility that TBK1- Δ C functions as a dominant negative form of the kinase?

5. The authors observe a TBK1-dependent recruitment of WIPI1 to Salmonella (Figure 1E), and for this reason they hypothesize that TBK1 catalytic activity/adaptor-binding capability affects Vps34 function. This finding would functionally connect a main player of the anti-bacterial autophagy, TBK1, with a main complex in the autophagy induction, the PI3KIII-complex. Such a link could be potentially relevant for the coordinated regulation of macroautophagy and anti-bacterial autophagy (a selective autophagy pathway) or could suggest that TBK1 triggers de novo autophagosome formation at Salmonella infection site. For this reason, I believe that the possible functional relationship between TBK1 and Vps34 is worthy to be further validated.

Regarding this, I would suggest to perform at least some of the following experiments: i) measure Vps34 activity in WT and TBK1^{-/-} MEFs, and in MEFs reconstituted with the kinase-dead TBK1 or TBK1- Δ C; ii) check for the localization of Vps34 and Vps34-interacting proteins at the Salmonella infection site, and check whether such an event could be dependent on TBK1 activity; iii) check for the TBK1-mediated enrichment in PI3P at Salmonella infection site by using PI3P binding probes (such as FYVE). This assay would assess whether TBK1 could selectively affect Vps34 activity at Salmonella infection site.

6. Figure 2A. The authors conclude that there is a difference in the Salmonella proliferation

restriction capability between TBK1- Δ C:Nap and TBK1- Δ C:Tank constructs. For this reason, I would suggest them to show in the figure the p-value specifically referring to these two samples. In the figures, it is not always clear to which samples the p-value refers to, and this should be better specified in the figure legend.

7. Thurston and colleagues show that TBK1 binding to galectin-8 and/or ubiquitin is necessary in order to restrain Salmonella hyperproliferation and to recruit WIPI1 to Salmonella close proximity. It would be of the highest interest to test whether these two events are actually functionally correlated. Is WIPI1 recruitment functional to TBK1-mediated prevention of Salmonella proliferation?

8. In some experiments, the different fusion constructs tested for the Salmonella proliferation restriction show very different expression levels (as an example, Figure 6B, showing very high expression levels of GFP and TBK1, and low levels of all the mutants). In this circumstance, the low expression levels of some of these constructs (as an example, constructs 5 and 6 in figure 6B) could also, at least partially, explain their incapability in restraining Salmonella proliferation. Similar expression levels among different constructs would be desirable in order to properly assess the functionality of the mutants.

9. In figure 6B it is shown that the depletion/mutation of a single Nemo ubiquitin binding domains is sufficient to prevent the capability of TBK1 Δ C:Nemo of restraining the Salmonella infection. Given that the replication curves associated with these constructs (TBK1 Δ C-Nemo Δ ZnF/D304N) are very close to TBK1 curve (used as a positive control), I would also report the p-values related to the difference between TBK1 and TBK1 Δ C-Nemo Δ ZnF/D304N.

Minor comments

1. The authors created and used a high number of different mutants in the manuscript, but most of them are poorly described. For this reason, I would add in the methods section a Table reporting all the constructs used in the manuscript, in which the deleted/fused/mutagenized amino acidic regions of the different proteins are reported per each construct. Also, where necessary, the proper references should be reported.

2. Page 6 of the manuscript. No reference is reported for the catalytically inactive mutant of TBK1 (TBK1 K38M).

3. Page 7 of the manuscript. I would rephrase "TBK1 controlling autophagy" as "TBK1 controlling anti-bacterial autophagy".

4. Two references are wrongly reported in the text in pages 6 and 8 of the manuscript.

5. WIPI1 is erroneously spelled (WIPI-1), page 8 of the manuscript.

6. Figure 3 is erroneously reported as Figure 3B in the text.

7. Fig. 4D. In order to help the reader, I would spell the acronyms (Hs, Bt, Gg, Xe) out in the figure legend. Out of curiosity, are these residues also conserved in *Mus musculus*?

EMBO Journal
Dr Andrea Leibfried

September 22nd, 2015

Dear Andrea,

Thank you for sending us the referee reports and giving us the opportunity to reply before taking an editorial decision.

We are glad that the referees consider the topic of our manuscript important and our work elegant. Most of the referee comments on existing data can be addressed easily.

Regarding new data, we agree with the referees that the next important step in understanding TBK1 function will be the identification of its downstream effector. Since the mechanism of TBK1 recruitment to cytosol-invading *Salmonella* comprises the main focus of our manuscript we consider a *de novo* screen to identify TBK1 effectors essential for anti-bacterial autophagy beyond the scope of the work reported here. However, many of the specific experiments suggested by the referees to address TBK1 effector functions are technically feasible and some may provide interesting insights. I am confident that by following the referee suggestions, we will provide further evidence to strengthen the manuscript with respect to the role of TBK1 in PI3P production and hence WIPI1 recruitment. In particular, we will test whether

- Optineurin and its phosphorylation by TBK1 are required for WIPI1 recruitment
- WIPI1 recruitment is mediated by PI3P and requires Vps34 activity
- Vps34 activity on bacteria requires TBK1 by imaging Vps34 activity with GFP-FYVE.

In addition I would like to point out that certain referee requests do not take into account the specifics of anti-bacterial autophagy. While in starvation-induced autophagy or drug-induced mitophagy large numbers of autophagosomes are formed in all cells, anti-bacterial autophagy generates single autophagosomes in only a fraction of cells. Bulk assays of enzymatic activity, as suggested by some of the referees, are therefore unlikely to yield results.

It would be most helpful if you could let us know your opinion on the chances of a manuscript that contained additional experiments as detailed in the point-to-point reply.

I am looking forward to hear from you.

With kind regards,

Felix

Referee 1#:

In this manuscript, Thurston, et al, investigate a mechanism by which the kinase TBK1 regulates autophagy. Binding of NDP52 to ubiquitin engaged TBK1, which was necessary for recruitment of autophagy regulating factors. The authors found that linking TBK1 directly to several different autophagy-related factors delineated specificity for downstream signaling components.

Overall, this work focuses on an important area of innate immune function that needs further elucidation, specifically how cytosolic bacteria trigger ubiquitin-mediated autophagy. In general, the manuscript is well written and the experiments executed with appropriate controls. The authors use elegant molecular biology approaches to identify critical downstream signaling mediators, and are nicely filling in the details of autophagy initiation. However, there are several major points of concern, indicated in more detail below. Additionally, although the authors do provide evidence for a more detailed hierarchy in initiating autophagy, the mechanism by which TBK1 recruits the upstream autophagy component, WIPI1, is hinted at, but still not addressed. These points, as well as the specific issues below, decreased my enthusiasm for the study in its present form.

Please see point 2 regarding WIPI1 recruitment.

Specific points the authors should address:

1. The magnitude of the phenotypes they showcase throughout this particular study are fairly modest, perhaps 1.5-3 fold, e.g., Fig. 2A. In their original paper (Thurston, Nat Imm 2009), the differences were much more striking. It was not evident why the phenotype quantitatively changed. In light of the concerns about the statistical analysis in point #4 below, this point should be addressed.

Thurston (Nat Immunol 2009) used human epithelial cells, this work is in murine TBK1-/- MEFs. Importantly, lack of TBK1 causes Salmonella hyperproliferation in both species, while quantitative differences are entirely expected given the differences in cell type and species.

2. The authors suggest several times in the paper that the way in which TBK1 controls bacterial proliferation is unknown. Data in Fig. 1 lead the authors to suggest that TBK1 kinase activity may regulate VPS34 function, leading to the observed differences in WIPI1 recruitment, which would be an exciting finding to address their rationale, but this is not tested.

To address the referee's suggestion we will test whether

- i) WIPI1 recruitment is mediated by PI3P and requires Vps34 activity
- ii) Vps34 activity on bacteria requires TBK1 by imaging Vps34 activity with GFP-FYVE
- iii) Optineurin and its phosphorylation by TBK1 are required for WIPI1 recruitment

3. In Fig. 5B, the data show that only 4% or so associate with GFP-WIPI1, which the authors propose as the TBK1-dependent determining factor in promoting anti-bacterial autophagy (as opposed to non-productive LC3 association with SCV). This number seems quite low for a mechanism proposed to restrict infection.

The low percentage of WIPI1+ bacteria most likely reflects transient association to the early autophagosome. We will test this hypothesis experimentally.

4. Statistical concerns: for example, in Fig. 6C - the authors again emphasize statistically significant differences, e.g., between 100 and 150 fold replication, which seems quite modest. In this figure and others, the figure legend indicates the data are "representative" of at least 2 repetitions of the experiment. From what I can understand from the figure legends, the statistics are done on triplicate cell samples and duplicate colony counts within the same experiment. If this is correct, this is not the appropriate way to do statistical analysis. For more

robust analysis, the statistical comparisons should be done with data across several experiments.

The referee interprets our description of the statistical analysis correctly.

Bacterial proliferation in epithelial cells depends on a multitude of factors, some of which are inducible and difficult to control precisely, such as Spi1, the bacterial type iii secretion system. As a result, levels of bacterial proliferation in control samples differ from day to day, thus preventing us from pooling data across several experiments.

Referee #2:

The study submitted by Thurston et al. investigates the role of TBK1 in mediating bacterial autophagy. The same group previously showed that NDP52 is a specific receptor for anti-bacterial autophagy that recruits TBK1 through specific adaptors, and recognizes bacteria through galectin-8 binding (Thurston et al. 2009, Nature Immunology and Thurston et al. 2012, Nature). This study expands further on the role of TBK1 while also investigating the structural basis for NDP52 binding to ubiquitin. The study is of high quality and the authors' conclusions are generally well supported by their results. Of particular interest is the observations that ubiquitin- or Gal8-binding by NDP52 are sufficient to mediate bacterial autophagy. However, some aspects have been neglected, such as the mechanism of action of TBK1 kinase activity, beyond the requirement for adaptor-mediated localization. Thus, while this manuscript certainly contains interesting findings and reinforces our understanding of bacterial autophagy, I believe it should be published in a more specialized journal, unless more mechanistic data is provided. Here are some specific points for the authors to consider:

1. Perhaps the biggest outstanding question, which has not been explored, is the mechanism of action of TBK1. What is the phosphorylation step required for TBK1-mediated WIPI1 recruitment? TBK1 phosphorylates Optineurin (OPTN) at Ser177, in both bacterial autophagy (Wild et al. 2011), as well as mitophagy (Lazarou et al. 2015). Is OPTN, as well as its phosphorylation, required for WIPI1 recruitment? Can a phosphomimetic OPTN bypass TBK1 recruitment for this process? Alternatively, the authors speculate that TBK1 might target the VPS34 complex, which would then make PI(3)P and recruit WIPI1. Another possibility is the potential involvement of TBK1 in the activation of the autophagy initiation complex (Atg1/ULK1 complex). These potential mechanisms can and should be tested experimentally.

The referee makes several straightforward suggestions.

Following his suggestions, we will test whether Optineurin is required for WIPI-1 recruitment to bacteria. If this turned out to be the case, we will test if expression of phospho-mimetic OPTN is sufficient to rescue WIPI1 recruitment in TBK1^{-/-} MEFs.

We will also follow the referee's advice to investigate PI(3)P production and the VPS34 complex in more detail:

- i) VPS34 activity needs to be measured in situ rather than biochemically in cell lysates, given the paucity of anti-bacterial autophagosomes. We will therefore deploy GFP-tagged FYVE domains, the standard PI(3)P receptor, to test whether PI(3)P is indeed produced in the vicinity of cytosol-invading bacteria.
- ii) We will test whether WIPI1 recruitment is indeed dependent on PI(3)P using mutants specifically deficient in binding to PI(3)P.

iii) We will attempt to visualize the VPS34 complex on cytosol-invading bacteria. The experiment will be technically difficult, given the association of VPS34 with many membranes. We will attempt to identify antibodies suitable for IF against ATG14, an autophagy specific subunit.

We however feel that analyzing the Atg1/ULK1 complex is beyond the scope for this manuscript. Analysis of this complex is the focus of another ongoing study from our laboratory.

2. The intent for testing exclusively LC3B recruitment is not clear (Figure 1D). NDP52 recruits LC3C; hence the recruitment of LC3C should also be tested in the context of TBK1 activity and function. Also, even though the recruitment of LC3B is normal in TBK1-null cells, an important factor to consider from the perspective of autophagy is to monitor the conversion of LC3 type I to type II forms. The authors also need to state the intent of this experiment more clearly: Rather than just saying 'visualize anti-bacterial autophagy attack', they should state whether they were expecting LC3B association to change and also provide the context for the use of LC3B as a marker with literature references.

We apologize for not making clearer why we only analyze LC3B. LC3C has been lost from the genome of mice and rats.

We do not think that investigating the conversion of LC3 type I into LC3 type II will give useful insights beyond what is already shown in the manuscript.

- i) Anti-bacterial autophagosomes are so rare compared to bulk autophagy that global measurements of LC3 lipidation are not informative.
- ii) Lipidation of LC3 is an absolute requirement for its recruitment to autophagosomes - since LC3 recruitment occurs normally in TBK1^{-/-} cells, analyzing its lipidation will not give further insights.

3. The authors fail to acknowledge a study published earlier this year about the role of NDP52 during the early and late stages of bacterial autophagy (Verlhac et al. 2015, Cell Host and Microbe). In particular, it was observed that mutation of C425 in the ZnF domain of NDP52 fails to restrict bacterial proliferation in cells treated with NDP52 siRNA, suggesting that ubiquitin-binding is not dispensable for bacterial autophagy. This contradicts the current study, which shows that the mutation D439K alone does not affect the function of TBK1 Δ C-NDP52. One major difference is that endogenous NDP52 is still present in the latter case, which may complement the activity of the chimeric construct through dimerization. The authors should eliminate the possibility that the observed effects may be mediated by endogenous NDP52, and should try to reconcile their observations with this other study or explain otherwise.

We apologize for not having included this study.

We disagree with the referee's comment that our data are in disagreement with Verlhac. The misunderstanding is caused by the entirely different questions asked in the two studies. Verlhac investigates the importance of ubiquitin for NDP52 function per se, while we investigate the recruitment of TBK1 as one of NDP52's many tasks. Therefore, while the overall function of NDP52 in antibacterial autophagy appears ubiquitin dependent, as demonstrated by Verlhac, recruitment of TBK1 is not. Verlhac and our study therefore provide complementary insight into anti-bacterial autophagy.

Regarding the '... possibility that the observed effects may be mediated by endogenous NDP52', mice and rat do not encode functional NDP52. Our experiments are therefore performed in cells carefully chosen for lack of TBK1 and NDP52.

4. In all experiments involving the fusion of TBK1 to different constructs of Nap1, NDP52, Gal8 or OPTN, control experiments should be performed with the cells transfected with only the respective inserts from Nap1, NDP52, Gal8 or OPTN, to eliminate the possibility that the

observed effects find their origin in the overexpression of those inserts.

We respectfully disagree with this request.

Our experiments were performed by complementing TBK1 deficient cells. Since complementation requires TBK1 activity (Fig1A), how could overexpression of Nap1, NDP52, Gal8, or OPTN possibly provide kinase activity? In addition we also compare the effects of TBK1 fused to WT and point mutants of NDP52, Gal8, or OPTN, which we feel constitute the most appropriate control for the experiments conducted.

Minor points:

5. In the description of the results for figure 1A, the authors conclude that "adaptor binding controls kinase activity spatially and/or temporarily". While I understand what the authors meant, it may be interpreted as if the adaptor protein controlled the kinase activity itself, for which there is no evidence. The sentence needs to be rephrased for clarity. Also, it's 'temporally', not 'temporarily'.

We will clarify this sentence as requested and thank the referee for identifying this typo.

6. Ubiquitination of S. Typhimurium was assessed after 1 and 4 hours (Figure 1C) while L3CB association was tested only 1 hour post-infection. Could a difference be observed 4 hour later? This needs to be justified especially since L3CB recruitment is a more 'downstream' process.

It has previously been reported that the peak of LC3 recruitment is at 1 h post infection and this is why we have analyzed this time point (Birmingham et al., 2006).

7. Figure 4 A, B and C: The authors should annotate the important residues on the figures, especially those that have been tested in assays. This is particularly essential for figure 4C in which chemical shift perturbations have been mapped and the interaction with ubiquitin is being shown. Besides the ubiquitin experts, no one can tell where the I44 patch is on the cartoon.

The requested changes will be made.

8. The manuscript needs to be checked for typos and labeling errors, such as "TKB1" on page 6. Also, there is no Figure 3B (p. 9). Finally, in Figure 6D, the labels should be "Tab2-ZnF" and "Mud1-UBA".

The requested changes will be made.

Referee #3:

The manuscript by Thurston et al. sheds new light on the mechanism of regulation of anti-bacterial autophagy, particularly focusing on an important player in this context, the kinase TBK1.

Indeed, the authors show that recruitment of TBK1 to cytosolic Salmonella is mediated, with similar efficiency, by different "eat-me signals", and that such a re-localization of TBK1 is fundamental to the prevention of Salmonella proliferation.

Interestingly, TBK1 presumably causes an enrichment in Phosphatidylinositol 3-phosphate

(PI3P) in the proximity of cytosolic bacteria, resulting into recruitment of WIPI1, an upstream regulator of the autophagy pathway. This finding opens an exciting scenario, suggesting an unexpected link between TBK1 and the upstream regulation of autophagy, possibly functional to efficiently restrict bacteria proliferation.

Altogether, the manuscript is well written, and the rationale, the experiment set-up and the interpretation of the data are thoughtful and elegant. Also, all along the manuscript, the authors take advantage of numerous deletion and single-point mutants, as well as TBK1-fusion proteins, in order to elegantly prove the relevance of TBK1 localization to *Salmonella* and to dissect the function of different "eat-me-signals" in this context. On the one hand, such an approach represents a great effort and is impressive, although, on the one other hand, in some cases the conclusions drawn by the authors exclusively rely on mutants that may need further characterization before being considered trustful and being used for functional assays.

We appreciate the referee's concern. Most point mutations are based on published structural information and we will provide a supplementary table with comprehensive information for every mutant.

This aspect represents one of my main concerns regarding the manuscript and I report below my detailed suggestions to further improve this study.

Major comments

1. Figure S1. The coomassie stain of purified GST proteins and the Western Blot of luciferase-NDP52 alleles, used in this experiment, are necessary controls. For this reason, they should be shown in order to support the reliability of the experiment itself.

Blots will be provided.

2. It is not clear to me why the authors did not test the binding capability of TBK1- Δ C construct for Sintbad and why did they keep this TBK1-binding partner out of their study.

We found that TBK1dC-Sintbad was unreliable expressed, most likely caused by Sintbad, given the large number of TBK1dC fusions successfully used in this study. We therefore focused on Nap1 that shares many properties with Sintbad in terms of association to *Salmonella* and binding to TBK1.

3. Figure 1B. The authors show that the restriction of *Salmonella* proliferation is prevented by TBK1 inhibition (MRT68843) in Atg5^{-/-} reconstituted MEFs. First, a reference should be cited supporting the efficiency and specificity of MRT68843 inhibitor, which should also be mentioned in the Methods section. Also, in order to further confirm the result obtained by using the inhibitor, the authors should also assess *Salmonella* proliferation in Atg5^{-/-} and reconstituted MEFs upon TBK1 knock-down.

We validated the efficiency of MRT68843 in inhibiting TBK1 by analyzing ISRE reporter activity controlled by IRF3, the canonical TBK1 substrate. MRT68843 inhibited TBK1 quantitatively even at a concentration as low as 10 nM (see Figure).

Whilst we could confirm our finding using TBK1 siRNA in Atg5^{-/-} and reconstituted MEFs we feel that this wouldn't add significantly to the data. We have however confirmed a non-additive affect of TBK1 inhibition in Atg5^{-/-} MEFs using another TBK1 inhibitor, MRT68601, which has been cited previously (Newman et al., 2012).

4. A truncated form of TBK1 (TBK1- Δ C) and its fusion proteins are extensively used all along the manuscript. As the authors properly tested, TBK1- Δ C is defective for binding to TBK1 adaptors (Figure S1). Nevertheless, the authors did not take into account the possibility that, messing up with TBK1 structure and localization, they could also accidentally affect TBK1 kinase activity, a possibility that is also clearly discussed in the literature (ref. in Helgason et al., 2013). On this line, the authors should check for the catalytic activity of TBK1- Δ C upon its over-expression in mammalian infected with Salmonella cells. Also, the phosphorylation levels of TBK1 substrates (such as p62, Optineurin, etc.) must be measured in TBK1^{-/-} MEFs reconstituted with WT, TBK1- Δ C and the kinase-dead TBK1.

Upon overexpression TBK1dC activates IRF3-dependent reporter activity, excluding the possibility of a severe kinase defect.

However, measuring TBK1dC activity in Salmonella infected cells is unlikely to give insight into TBK1 activation against cytosol-invading bacteria since the majority of TBK1 activity will be induced by TLRs, not by rare cytosol-invading bacteria.

Similarly, phosphorylation levels of TBK1 substrates (p62, OPTN) will reflect TLR induced TBK1 activity, not TBK1 activity from anti-bacterial autophagy.

Finally, given that TBK1 has been reported to dimerize, did the author test the possibility that TBK1- Δ C functions as a dominant negative form of the kinase?

As we complement a TBK1 KO MEF line, TBK1dC cannot be functioning in a dominant negative capacity in our assays.

5. The authors observe a TBK1-dependent recruitment of WIPI1 to Salmonella (Figure 1E), and for this reason they hypothesize that TBK1 catalytic activity/adaptor-binding capability affects Vps34 function. This finding would functionally connect a main player of the anti-bacterial autophagy, TBK1, with a main complex in the autophagy induction, the PI3KIII-complex. Such a link could be potentially relevant for the coordinated regulation of macroautophagy and anti-bacterial autophagy (a selective autophagy pathway) or could suggest that TBK1 triggers de novo autophagosome formation at Salmonella infection site. For this reason, I believe that the possible functional relationship between TBK1 and Vps34 is worthy to be further validated.

Regarding this, I would suggest to perform at least some of the following experiments:

- i) measure Vps34 activity in WT and TBK1-/- MEFs, and in MEFs reconstituted with the kinase-dead TBK1 or TBK1- Δ C;
- ii) check for the localization of Vps34 and Vps34-interacting proteins at the Salmonella infection site, and check whether such an event could be dependent on TBK1 activity;
- iii) check for the TBK1-mediated enrichment in PI3P at Salmonella infection site by using PI3P binding probes (such as FYVE). This assay would assess whether TBK1 could selectively affect Vps34 activity at Salmonella infection site.

i – Measuring total Vps34 activity will not be insightful, given the low abundance of anti-bacterial autophagosomes and the requirement for Vps34 activity in other constitutive cellular pathways. However, we will attempt to measure Vps34 activity on invading bacteria using GFP-tagged FYFE domains.

ii – We will investigate the localization of Vps34 as suggested by the referee.

iii- We will investigate recruitment of GFP-FYFE as a readout of localized Vps34 activity. We also will investigate whether WIPI1 recruitment is PI3P dependent using binding deficient mutants.

6. Figure 2A. The authors conclude that there is a difference in the Salmonella proliferation restriction capability between TBK1- Δ C: Nap and TBK1- Δ C: Tank constructs. For this reason, I would suggest them to show in the figure the p-value specifically referring to these two samples. In the figures, it is not always clear to which samples the p-value refers to, and this should be better specified in the figure legend.

Re-analysis of the data revealed a significant difference between the TBK1- Δ C: Nap and TBK1- Δ C: Tank constructs (two-tailed t-test, $p < 0.01$).

7. Thurston and colleagues show that TBK1 binding to galectin-8 and/or ubiquitin is necessary in order to restrain Salmonella hyperproliferation and to recruit WIPI1 to Salmonella close proximity. It would be of the highest interest to test whether these two events are actually functionally correlated. Is WIPI1 recruitment functional to TBK1-mediated prevention of Salmonella proliferation?

Knockdowns of WIPI1 could be attempted. However, the existence of many (!) isoforms and multiple family members make this an unlikely achievement.

8. In some experiments, the different fusion constructs tested for the Salmonella proliferation restriction show very different expression levels (as an example, Figure 6B, showing very high expression levels of GFP and TBK1, and low levels of all the mutants). In this circumstance, the low expression levels of some of these constructs (as an example, constructs 5 and 6 in figure 6B) could also, at least partially, explain their incapability in restraining Salmonella proliferation. Similar expression levels among different constructs would be desirable in order to properly assess the functionality of the mutants.

We agree that similar expression levels would be desirable. However, since all transgenes were introduced by transduction at very low MOI (i.e. less than one virus per cell) followed by drug selection, the expression levels are inherent to the fusion proteins and do not represent experimental artefacts.

Importantly, expression levels do not correlate with functional complementation. For example, in Figure 6B construct 5 is expressed to a higher level than construct 4, which complements, whereas 5 does not.

9. In figure 6B it is shown that the depletion/mutation of a single Nemo ubiquitin binding domains is sufficient to prevent the capability of TBK1 Δ C:Nemo of restraining the Salmonella infection. Given that the replication curves associated with these constructs (TBK1 Δ C-Nemo Δ ZnF/D304N) are very close to TBK1 curve (used as a positive control), I would also report the p-values related to the difference between TBK1 and TBK1 Δ C-Nemo Δ ZnF/D304N.

Reanalysis revealed a significant difference between TBK1 and TBK1 Δ C-Nemo Δ ZnF ($p < 0.05$) as well as TBK1 Δ C-NemoD304N ($p < 0.01$).

Minor comments

1. The authors created and used a high number of different mutants in the manuscript, but most of them are poorly described. For this reason, I would add in the methods section a Table reporting all the constructs used in the manuscript, in which the deleted/fused/mutagenized amino acidic regions of the different proteins are reported per each construct. Also, where necessary, the proper references should be reported.

Data will be added.

2. Page 6 of the manuscript. No reference is reported for the catalytically inactive mutant of TBK1 (TBK1 K38M).

Pomerantz and Baltimore 1999.

3. Page 7 of the manuscript. I would rephrase "TBK1 controlling autophagy" as "TBK1 controlling anti-bacterial autophagy".

Suggested change will be included.

4. Two references are wrongly reported in the text in pages 6 and 8 of the manuscript.

We will correct the references. Could the referee specify the references, please?

5. WIPI1 is erroneously spelled (WIPI-1), page 8 of the manuscript.

Error will be corrected

6. Figure 3 is erroneously reported as Figure 3B in the text.

Error will be corrected.

7. Fig. 4D. In order to help the reader, I would spell the acronyms (Hs, Bt, Gg, Xe) out in the figure legend. Out of curiosity, are these residues also conserved in *Mus musculus*?

Change will be implemented.

Mice and rats do not encode full length NDP52.

Thank you very much for sending me your point-by-point response to the referees' comments upfront. I appreciate your outline and found it very helpful and productive.

I think the proposed additional experiments go into the right direction. If the additional experiments that address the link between TBK1 and OPTN, WIPI1, and Vps34 provide more definitive insight into the function of TBK1 in this context, I'd be happy to further consider your manuscript for publication here. However, the outcome of the proposed experiments are of course hard to predict. Therefore, I am sorry that I have no other choice but to reject your manuscript at this stage.

Should you be able to strengthen the link between TBK1 and the autophagic machinery as outlined, I'd be however happy to look at your manuscript again, and I would also engage with the same set of referees to avoid any additional revisions. Please note that I would have to first check for novelty upon re-submission though.

Referee 1#:

In this manuscript, Thurston, et al, investigate a mechanism by which the kinase TBK1 regulates autophagy. Binding of NDP52 to ubiquitin engaged TBK1, which was necessary for recruitment of autophagy regulating factors. The authors found that linking TBK1 directly to several different autophagy-related factors delineated specificity for downstream signaling components.

Overall, this work focuses on an important area of innate immune function that needs further elucidation, specifically how cytosolic bacteria trigger ubiquitin-mediated autophagy. In general, the manuscript is well written and the experiments executed with appropriate controls. The authors use elegant molecular biology approaches to identify critical downstream signaling mediators, and are nicely filling in the details of autophagy initiation. However, there are several major points of concern, indicated in more detail below. Additionally, although the authors do provide evidence for a more detailed hierarchy in initiating autophagy, the mechanism by which TBK1 recruits the upstream autophagy component, WIPI1, is hinted at, but still not addressed. These points, as well as the specific issues below, decreased my enthusiasm for the study in its present form.

We thank the reviewer for summarizing our findings concisely and for appreciating the importance of the question under study. We followed the referee's suggestion to deepen the investigation of WIPI proteins. Details are given below.

Specific points the authors should address:

1. The magnitude of the phenotypes they showcase throughout this particular study are fairly modest, perhaps 1.5-3 fold, e.g., Fig. 2A. In their original paper (Thurston, Nat Imm 2009), the differences were much more striking. It was not evident why the phenotype quantitatively changed. In light of the concerns about the statistical analysis in point #4 below, this point should be addressed.

Thurston (Nat Immunol 2009) used human epithelial cells, this work is in murine TBK1^{-/-} MEFs. Importantly, lack of TBK1 causes Salmonella hyperproliferation in either species, while quantitative differences are entirely expected given the differences in cell type and species.

2. The authors suggest several times in the paper that the way in which TBK1 controls bacterial proliferation is unknown. Data in Fig. 1 lead the authors to suggest that TBK1 kinase activity may regulate VPS34 function, leading to the observed differences in WIPI1 recruitment, which would be an exciting finding to address their rationale, but this is not tested.

New panels added to address this question: 2B, 2C, 2D, 2E, 2F, 2G, 2H, S2A, S2B, 3A, 3B.

To address the reviewer's point we extended our study to the whole WIPI family. We found that both WIPI1 and WIPI2B, unlike WIPI3 and WIPI4, localized to Salmonella in a TBK1-dependent manner

(**new Fig2B**). Importantly, this association is abrogated by the addition of wortmannin and requires functional PI3P-binding sites in WIPIs (**new Fig2B**). Endogenous WIPI2 also associates with *Salmonella*, again in a TBK1 and PI(3)P-dependent manner (**new Fig.2C, new Fig.S2B**). However, in contrast to WIPIs, recruitment of another PI(3)P-binding protein, DFCP1, did not require TBK1 despite also being sensitive to wortmannin treatment and mutational inactivation of its PI(3)P-binding site (**new Fig.2D,E,F**). Data on DFCP1 clearly indicate that TBK1 does not regulate VPS34 activity. We therefore conclude that TBK1 is not acting via VPS34 but rather that TBK1 and VPS34 independently control the recruitment of WIPI1 and WIPI2 to *S. Typhimurium*.

Given the dependence of WIPI recruitment on VPS34, we analyzed whether TBK1 controls the occurrence PI(3,5)P₂, a PI(3)P-derived ligand for ATG18 in yeast. However, the accumulation of GFP:MLIN*2, a PI(3,5)P₂-specific probe, was not dependent on TBK1 (**new Fig.2G,H**).

Alternatively, instead of affecting WIPI recruitment by altering PI(3)P or PI(3,5)P₂ levels, TBK1 could directly bind WIPI proteins. However, a LUMIER-based immunoprecipitation assay revealed no such interaction (**data below**).

Given the lack of WIPI-binding to TBK1 we tested whether WIPI recruitment depends on the phosphorylation of optineurin, the only known TBK1 substrate in anti-bacterial autophagy (Wild et al, 2011). Cells lacking Optineurin recruited WIPI1 and WIPI2B normally to *S. Typhimurium*, suggesting that phosphorylation of a substrate other than Optineurin is essential for WIPI1/2 recruitment in anti-bacterial autophagy (**new Fig.S2C**). Finally we tested the interdependence of WIPI1 and WIPI2B recruitment and found that neither protein was required for the recruitment of the other (**Fig.S2D**).

3. In Fig. 5B, the data show that only 4% or so associate with GFP-WIPI1, which the authors propose as the TBK1-dependent determining factor in promoting anti-bacterial autophagy (as opposed to non-productive LC3 association with SCV). This number seems quite low for a mechanism proposed to restrict infection.

The low percentage of WIPI+ bacteria reflects the transient association of WIPI with autophagosomes. Transient WIPI recruitment is in striking contrast to the long-lasting association of bacteria with LC3. See this movie for an illustration

<http://tinyurl.com/zak55a9>

Salmonella (blue), LC3 (red), WIPI (green)

4. Statistical concerns: for example, in Fig. 6C - the authors again emphasize statistically significant differences, e.g., between 100 and 150 fold replication, which seems quite modest. In this figure and others, the figure legend indicates the data are "representative" of at least 2 repetitions of the experiment. From what I can understand from the figure legends, the statistics are done on triplicate cell samples and duplicate colony counts within the same experiment. If this is correct, this is not the appropriate way to do statistical analysis. For more robust analysis, the statistical comparisons should be done with data across several experiments.

We appreciate the referee's suggestion for an alternative statistical analysis across multiple experiments. Results from pooled experiments completely verified the conclusions we had drawn from representative experiments.

New Fig.S1A – related to Fig1A. Only TBK1 significantly reduces bacterial replication.

New Fig.S3A – related to Fig4A. TBK1, TBK1dC-Nap1 and TBK1dC-Nap1 N85 significantly reduce bacterial replication.

New Fig.S3B – related to Fig4D and 5F. TBK1, TBK1dC-NDP52, TBK1dC-NDP52dSKICH, TBK1dC-NDP52 L374A and TBK1dC-NDP52 D439K significantly reduce bacterial replication.

New Fig.S7A – related to Fig6A. TBK1 and TBK1dC-gal8 significantly reduce bacterial replication.

New Fig.S7B – related to Fig7B. TBK1 and TBK1dC-Nemo significantly reduce bacterial replication.

New Fig.S7C – related to Fig7C. TBK1, TBK1dC-Optn and TBK1dC-Optn dLIR significantly reduce bacterial replication.

New Fig.S7D – related to Fig7D. TBK1, TBK1dC-TAB2-NZF and TBK1dC-mud1-UBA significantly reduce bacterial replication.

Referee #2:

The study submitted by Thurston et al. investigates the role of TBK1 in mediating bacterial autophagy. The same group previously showed that NDP52 is a specific receptor for anti-bacterial autophagy that recruits TBK1 through specific adaptors, and recognizes bacteria through galectin-8 binding (Thurston et al. 2009, Nature Immunology and Thurston et al. 2012, Nature). This study expands further on the role of TBK1 while also investigating the structural basis for NDP52 binding to ubiquitin. The study is of high quality and the authors' conclusions are generally well supported by their results. Of particular interest is the observations that ubiquitin- or Gal8-binding by NDP52 are sufficient to mediate bacterial autophagy. However, some aspects have been neglected, such as the mechanism of action of TBK1 kinase activity, beyond the requirement for adaptor-mediated localization. Thus, while this manuscript certainly contains interesting findings and reinforces our understanding of bacterial autophagy, I believe it should be published in a more specialized journal, unless more mechanistic data is provided. Here are some specific points for the authors to consider:

1. Perhaps the biggest outstanding question, which has not been explored, is the mechanism of action of TBK1. What is the phosphorylation step required for TBK1-mediated WIPI1 recruitment? TBK1 phosphorylates Optineurin (OPTN) at Ser177, in both bacterial autophagy (Wild et al. 2011), as well as mitophagy (Lazarou et al. 2015). Is OPTN, as well as its phosphorylation, required for WIPI1 recruitment? Can a phosphomimetic OPTN bypass TBK1 recruitment for this process? Alternatively, the authors speculate that TBK1 might target the VPS34 complex, which would then make PI(3)P and recruit WIPI1. Another possibility is the potential involvement of TBK1 in the activation of the autophagy initiation complex (Atg1/ULK1 complex). These potential mechanisms can and should be tested experimentally.

We designed experiments to test several of the referee's straightforward hypotheses.

- *Ref: Is OPTN, as well as its phosphorylation, required for WIPI1 recruitment? Optineurin per se is not required for WIPI association with Salmonella (new FigS2C), therefore its phosphorylation cannot be required either.*

- *Ref: Can a phosphomimetic OPTN bypass TBK1 recruitment for this process? Phosphomimetic Optn alleles are insufficient to rescue control of bacterial replication in TBK1-/- MEFs (data below, A), despite expression (data below, B) and association to Salmonella (data below, C).*

- *Ref: Is TBK1 targeting the VPS34 complex? New panels added to address this question: 2B, 2C, 2D, 2E, 2F, 2G, 2H, S2A, S2B, 3A, 3B.*

The short answer is that *TBK1* and *VPS34* are independently required to recruit *WIPI1* and *WIPI2* to bacteria. Please see below for the experiments that led us to this conclusion.

To address the reviewer's point we extended our study to the whole *WIPI* family. We found that both *WIPI1* and *WIPI2B*, unlike *WIPI3* and *WIPI4*, localized to *Salmonella* in a *TBK1*-dependent manner (new Fig2B). Importantly, this association is abrogated by the addition of wortmannin and requires functional *PI3P*-binding sites in *WIPIs* (new Fig2B). Endogenous *WIPI2* also associates with *Salmonella*, again in a *TBK1* and *PI3P*-dependent manner (new Fig.2C, new Fig.S2). However, in contrast to *WIPIs*, recruitment of another *PI3P*-binding protein, *DFCP1*, did not require *TBK1* despite also being sensitive to wortmannin treatment and mutational inactivation of its *PI(3)P*-binding site (new Fig.2D,E,F). Data on *DFCP1* clearly indicate that *TBK1* does not regulate

VPS34 activity. We therefore conclude that TBK1 is not acting via VPS34 but rather that TBK1 and VPS34 independently control the recruitment of WIP11 and WIP12 to S. Typhimurium.

*Given the dependence of WIP1 recruitment on VPS34, we analyzed whether TBK1 controls the occurrence PI(3,5)P₂, a PI(3)P-derived ligand for ATG18 in yeast. However, the accumulation of GFP:MLIN*2, a PI(3,5)P₂-specific probe, was not dependent on TBK1 (new Fig.2G,H).*

Alternatively, instead of affecting WIP1 recruitment by altering PI3P or PI(3,5)P levels, TBK1 could directly bind WIP1 proteins. However, a LUMIER assay revealed no such interaction (data below).

Given the lack of WIP1-binding to TBK1 we tested whether WIP1 recruitment depends on the phosphorylation of optineurin, the only known TBK1 substrate in anti-bacterial autophagy (Wild et al, 2011). Cells lacking Optineurin recruited WIP11 and WIP12B normally to S. Typhimurium, suggesting that phosphorylation of a substrate other than Optineurin is essential for WIP11/2 recruitment in anti-bacterial autophagy (new Fig.S2C). Finally we tested the interdependence of WIP11 and WIP12B recruitment and found that neither protein was required for the recruitment of the other (Fig.S2D).

2. The intent for testing exclusively LC3B recruitment is not clear (Figure 1D). NDP52 recruits LC3C; hence the recruitment of LC3C should also be tested in the context of TBK1 activity and function. Also, even though the recruitment of LC3B is normal in TBK1-null cells, an important factor to consider from the perspective of autophagy is to monitor the conversion of LC3 type I to type II forms. The authors also need to state the intent of this experiment more clearly: Rather than just saying 'visualize anti-bacterial autophagy attack', they should state whether they were expecting LC3B association to change and also provide the context for the use of LC3B as a marker with literature references.

We apologize for not making clear why we only analyze LC3B. LC3C has been lost from the genome of mice and rats.

We respectfully disagree with the referee that regarding the importance of investigating the conversion of LC3 type I into LC3 type II for two reasons:

- i) Anti-bacterial autophagosomes are so rare compared to bulk autophagy that global measurements of LC3 lipidation are not informative.*
- ii) Lipidation of LC3 is an absolute requirement for its recruitment to autophagosomes - since LC3 recruitment occurs normally in TBK1^{-/-} cells, analyzing its lipidation will not give further insights.*

3. The authors fail to acknowledge a study published earlier this year about the role of NDP52 during the early and late stages of bacterial autophagy (Verlhac et al. 2015, Cell Host and Microbe). In particular, it was observed that mutation of C425 in the ZnF domain of NDP52 fails to restrict bacterial proliferation in cells treated with NDP52 siRNA, suggesting that ubiquitin-binding is not dispensable for bacterial autophagy. This contradicts the current study, which shows that the mutation D439K alone does not affect of the function of TBK1 Δ C-NDP52. One major difference is that endogenous NDP52 is still present in the latter case, which may complement the activity of the chimeric construct through dimerization. The authors should eliminate the possibility that the observed effects may be mediated by endogenous NDP52, and should try to reconcile their observations with this other study or explain otherwise.

We apologize for not having had this study included

However, we respectfully disagree with the referee's comment that our data contradict the study by Verlhac et al. The misunderstanding is caused by the entirely different questions asked in the two studies. Verlhac investigates the importance of ubiquitin for NDP52 function per se, while we investigate the recruitment of TBK1 as one of NDP52's many tasks. Therefore, while the overall function of NDP52 in antibacterial autophagy appears ubiquitin dependent, as demonstrated by Verlhac, recruitment of TBK1 Δ C-NDP52 is not. Verlhac and our study therefore provide complementary insight into anti-bacterial autophagy.

Regarding the '... possibility that the observed effects may be mediated by endogenous NDP52', mice and rat do not encode functional NDP52. Our experiments are therefore performed in cells carefully chosen for lack of TBK1 and NDP52.

4. In all experiments involving the fusion of TBK1 to different constructs of Nap1, NDP52, Gal8 or OPTN, control experiments should be performed with the cells transfected with only the respective inserts from Nap1, NDP52, Gal8 or OPTN, to eliminate the possibility that the observed effects find their origin in the overexpression of those inserts.

Our experiments were performed by complementing TBK1 deficient cells. Since complementation requires TBK1 activity (Fig1A), how could overexpression of Nap1, NDP52, Gal8, or OPTN possibly provide kinase activity? We rather think that the most appropriate control for our experiments is the fusion of TBK1 to WT and point mutants of NDP52, Gal8 or Optn.

Minor points:

5. In the description of the results for figure 1A, the authors conclude that "adaptor binding controls kinase activity spatially and/or temporarily". While I understand what the authors meant, it may be interpreted as if the adaptor protein controlled the kinase activity itself, for which there is no evidence. The sentence needs to be rephrased for clarity. Also, it's 'temporally', not 'temporarily'.

We thank the referee for pointing out a potentially confusing phrase. The sentence now reads "We therefore conclude that the catalytic activity of TBK1 and its ability to bind adaptor proteins are equally important to protect cells against S. Typhimurium, most likely because adaptor binding controls TBK1 spatially and / or temporally."

6. Ubiquitination of S. Typhimurium was assessed after 1 and 4 hours (Figure 1C) while L3CB association was tested only 1 hour post-infection. Could a difference be observed 4 hour later? This needs to be justified especially since L3CB recruitment is a more 'downstream' process.

It has previously been reported that the peak of LC3 recruitment is at 1 h post infection and this is why we have analyzed this time point (Birmingham et al., 2006).

7. Figure 4 A, B and C: The authors should annotate the important residues on the figures, especially those that have been tested in assays. This is particularly essential for figure 4C in which chemical shift perturbations have been mapped and the interaction with ubiquitin is being shown. Besides the ubiquitin experts, no one can tell where the I44 patch is on the cartoon.

The requested changes have been made (now Figure 5C).

8. The manuscript needs to be checked for typos and labeling errors, such as "TKB1" on page 6. Also, there is no Figure 3B (p. 9). Finally, in Figure 6D, the labels should be "Tab2-ZnF" and "Mud1-UBA".

The requested changes have been made.

Referee #3:

The manuscript by Thurston et al. sheds new light on the mechanism of regulation of anti-bacterial autophagy, particularly focusing on an important player in this context, the kinase TBK1. Indeed, the authors show that recruitment of TBK1 to cytosolic Salmonella is mediated, with similar efficiency, by different "eat-me signals", and that such a re-localization of TBK1 is fundamental to the prevention of Salmonella proliferation.

Interestingly, TBK1 presumably causes an enrichment in Phosphatidylinositol 3-phosphate (PI3P) in the proximity of cytosolic bacteria, resulting into recruitment of WIPI1, an upstream regulator of the autophagy pathway. This finding opens an exciting scenario, suggesting an unexpected link between TBK1 and the upstream regulation of autophagy, possibly functional to efficiently restrict bacteria proliferation.

Altogether, the manuscript is well written, and the rationale, the experiment set-up and the interpretation of the data are thoughtful and elegant. Also, all along the manuscript, the authors take advantage of numerous deletion and single-point mutants, as well as TBK1-fusion proteins, in order to elegantly prove the relevance of TBK1 localization to Salmonella and to dissect the function of different "eat-me-signals" in this context. On the one hand, such an approach represents a great effort and is impressive, although, on the one other hand, in some cases the conclusions drawn by the authors exclusively rely on mutants that may need further characterization before being considered trustful and being used for functional assays. This aspect represents one of my main concerns regarding the manuscript and I report below my detailed suggestions to further improve this study.

We are glad to read that the referee considers our study elegant and the link between TBK1 and the upstream autophagy machinery exciting.

We also appreciate the referee's concern regarding the many point mutants used in our study. Most point mutations used in this study are based on published structural information and we now provide a table with comprehensive information for every mutant.

Major comments

1. Figure S1. The coomassie stain of purified GST proteins and the Western Blot of luciferase-NDP52 alleles, used in this experiment, are necessary controls. For this reason, they should be shown in order to support the reliability of the experiment itself.

The requested blots have been added.

(We also noticed mistakes in the legend for Figure S1, which have been corrected in the current version.)

2. It is not clear to me why the authors did not test the binding capability of TBK1- Δ C construct for Sintbad and why did they keep this TBK1-binding partner out of their study.

We found that TBK1 Δ C-Sintbad was unreliably expressed, most likely caused by Sintbad, given the large number of TBK1 Δ C fusions successfully used in this study. We therefore focused on Nap1 that shares many properties with Sintbad in terms of association to Salmonella and binding to TBK1.

3. Figure 1B. The authors show that the restriction of Salmonella proliferation is prevented by TBK1 inhibition (MRT68843) in Atg5^{-/-} reconstituted MEFs. First, a reference should be cited supporting the efficiency and specificity of MRT68843 inhibitor, which should also be mentioned in the Methods section. Also, in order to further confirm the result obtained by using the inhibitor, the authors should also assess Salmonella proliferation in Atg5^{-/-} and reconstituted MEFs upon TBK1 knock-down.

We validated the efficiency of MRT68843 in inhibiting TBK1 by analyzing ISRE reporter activity controlled by IRF3, the canonical TBK1 substrate. MRT68843 inhibited TBK1 quantitatively at a concentration as low as 10 nM (new FigS1D).

4. A truncated form of TBK1 (TBK1-ΔC) and its fusion proteins are extensively used all along the manuscript. As the authors properly tested, TBK1-ΔC is defective for binding to TBK1 adaptors (Figure S1). Nevertheless, the authors did not take into account the possibility that, messing up with TBK1 structure and localization, they could also accidentally affect TBK1 kinase activity, a possibility that is also clearly discussed in the literature (ref. in Helgason et al., 2013). On this line, the authors should check for the catalytic activity of TBK1-ΔC upon its over-expression in mammalian infected with Salmonella cells. Also, the phosphorylation levels of TBK1 substrates (such as p62, Optineurin, etc.) must be measured in TBK1^{-/-} MEFs reconstituted with WT, TBK1-ΔC and the kinase-dead TBK1.

Upon overexpression of TBK1ΔC activates IRF3-dependent reporter activity, excluding the possibility of a severe kinase defect (new FigS1C).

The biochemical assays suggested by the referee are technically difficult, given the low frequency of cytosol-invading bacteria compared to the large excess of extracellular bacteria during the infection period. Directly measuring TBK1ΔC activity in Salmonella infected cells is therefore unlikely to give insight into TBK1 activation against cytosol-invading bacteria since the majority of TBK1 activity will be induced by TLRs, not by rare cytosol-invading bacteria. Similarly, global phosphorylation levels of TBK1 substrates (p62, OPTN) will reflect TLR induced TBK1 activity, not TBK1 activity related to anti-bacterial autophagy.

Finally, given that TBK1 has been reported to dimerize, did the author test the possibility that TBK1-ΔC functions as a dominant negative form of the kinase?

Since we complement TBK1^{-/-} cells, TBK1ΔC cannot function in a dominant negative capacity in our assays.

5. The authors observe a TBK1-dependent recruitment of WIPI1 to Salmonella (Figure 1E), and for this reason they hypothesize that TBK1 catalytic activity/adaptor-binding capability affects Vps34 function. This finding would functionally connect a main player of the anti-bacterial autophagy, TBK1, with a main complex in the autophagy induction, the PI3KIII-complex. Such a link could be potentially relevant for the coordinated regulation of macroautophagy and anti-bacterial autophagy (a selective autophagy pathway) or could suggest that TBK1 triggers de novo autophagosome formation at Salmonella infection site. For this reason, I believe that the possible functional relationship between TBK1 and Vps34 is worthy to be further validated.

Regarding this, I would suggest to perform at least some of the following experiments:

- i) measure Vps34 activity in WT and TBK1^{-/-} MEFs, and in MEFs reconstituted with the kinase-dead TBK1 or TBK1-ΔC;
- ii) check for the localization of Vps34 and Vps34-interacting proteins at the Salmonella infection site, and check whether such an event could be dependent on TBK1 activity;
- iii) check for the TBK1-mediated enrichment in PI3P at Salmonella infection site by using PI3P binding probes (such as FYVE). This assay would assess whether TBK1 could selectively affect Vps34 activity at Salmonella infection site.

We appreciate the referee's suggestions, which we addressed experimentally. New panels added: 2B, 2C, 2D, 2E, 2F, 2G, 2H, S2A, S2B, 3A, 3B.

To address the reviewer's point we extended our study to the whole WIPI family. We found that both WIPI1 and WIPI2B, unlike WIPI3 and WIPI4, localized to Salmonella in a TBK1-dependent manner (**new Fig2B**). Importantly, this association is abrogated by the addition of wortmannin and requires functional PI3P-binding sites in WIPIs (**new Fig2B**). Endogenous WIPI2 also associates with Salmonella, again in a TBK1 and PI(3)P-dependent manner (**new Fig.2C, new Fig.S2**). However, in contrast to WIPIs, recruitment of another PI3P-binding protein, DFCP1, did not require TBK1 despite also being sensitive to wortmannin treatment and mutational inactivation of its PI(3)P-binding site (**new Fig.2D,E,F**). Data on DFCP1 clearly indicate that TBK1 does not regulate VPS34 activity. We therefore conclude that TBK1 is not acting via VPS34 but rather that TBK1 and VPS34 independently control the recruitment of WIPI1 and WIPI2 to *S. Typhimurium*.

Given the dependence of WIPI recruitment on VPS34, we analyzed whether TBK1 controls the occurrence PI(3,5)P₂, a PI(3)P-derived ligand for ATG18 in yeast. However, the accumulation of GFP:MLN*2, a PI(3,5)P₂-specific probe, was not dependent on TBK1 (**new Fig.2G,H**).

Alternatively, instead of affecting WIPI recruitment by altering PI(3)P or PI(3,5)P₂ levels, TBK1 could directly bind WIPI proteins. However, a LUMIER assay revealed no such interaction (**data below**).

Given the lack of WIPI-binding to TBK1 we tested whether WIPI recruitment depends on the phosphorylation of optineurin, the only known TBK1 substrate in anti-bacterial autophagy (Wild et al, 2011). Cells lacking Optineurin recruited WIPI1 and WIPI2B normally to *S. Typhimurium*, suggesting that phosphorylation of a substrate other than Optineurin is essential for WIPI1/2 recruitment in anti-bacterial autophagy (**new Fig.S2C**). Finally we tested the interdependence of WIPI1 and WIPI2B recruitment and found that neither protein was required for the recruitment of the other (**Fig.S2D**).

6. Figure 2A. The authors conclude that there is a difference in the Salmonella proliferation restriction capability between TBK1-ΔC:Nap and TBK1-ΔC:Tank constructs. For this reason, I would suggest them to show in the figure the p-value specifically referring to these two samples. In the figures, it is not always clear to which samples the p-value refers to, and this should be better specified in the figure legend.

Re-analysis of the data revealed a significant difference between the TBK1-ΔC:Nap and TBK1-ΔC:Tank constructs (two-tailed t-test, $p < 0.01$ (Fig.4A) or one-way ANOVA $p < 0.05$ (Fig.S3A)). Legends have been modified as requested.

7. Thurston and colleagues show that TBK1 binding to galectin-8 and/or ubiquitin is necessary in order to restrain Salmonella hyperproliferation and to recruit WIPI1 to Salmonella close proximity. It would be of the highest interest to test whether these two events are actually functionally correlated. Is WIPI1 recruitment functional to TBK1-mediated prevention of Salmonella proliferation?

We have extended our study to include the whole WIPI family. WIPI1 and 2B, in contrast to WIPI3 and 4, are recruited to S. Typhimurium (new Fig.2B).

Recruitment of WIPI1 and WIPI2B is not interdependent (new Fig.S2D).

WIPI2 restricts proliferation of S. Typhimurium, WIPI1 is not essential (new Fig.3A,B).

8. In some experiments, the different fusion constructs tested for the Salmonella proliferation restriction show very different expression levels (as an example, Figure 6B, showing very high expression levels of GFP and TBK1, and low levels of all the mutants). In this circumstance, the low expression levels of some of these constructs (as an example, constructs 5 and 6 in figure 6B) could also, at least partially, explain their incapability in restraining Salmonella proliferation. Similar expression levels among different constructs would be desirable in order to properly assess the functionality of the mutants.

We agree that similar expression levels would be desirable. However, since all transgenes were introduced by transduction at very low MOI (i.e. less than one virus per cell) followed by drug selection, the expression levels are inherent to the fusion proteins and do not represent experimental artifacts.

Importantly, expression levels do not correlate with functional complementation. For example, in Figure 7B constructs 3 and 5 are expressed to a higher level than construct 4, which complements, whereas 3 and 5 do not.

9. In figure 6B it is shown that the depletion/mutation of a single Nemo ubiquitin binding domains is sufficient to prevent the capability of TBK1 Δ C:Nemo of restraining the Salmonella infection. Given that the replication curves associated with these constructs (TBK1 Δ C-Nemo Δ ZnF/D304N) are very close to TBK1 curve (used as a positive control), I would also report the p-values related to the difference between TBK1 and TBK1 Δ C-Nemo Δ ZnF/D304N.

Reanalysis of data from multiple experiments (new Fig.S7B) confirmed that only TBK1 and TBK1 Δ C-Nemo controlled bacterial replication when compared to TBK1 Δ C expressing cells.

Minor comments

1. The authors created and used a high number of different mutants in the manuscript, but most of them are poorly described. For this reason, I would add in the methods section a Table reporting all the constructs used in the manuscript, in which the deleted/fused/mutagenized amino acidic regions of the different proteins are reported per each construct. Also, where necessary, the proper references should be reported.

A table with construct names, descriptions and references has been added to the Materials and Methods section.

2. Page 6 of the manuscript. No reference is reported for the catalytically inactive mutant of TBK1 (TBK1 K38M).

The reference has been added (Pomerantz and Baltimore 1999).

3. Page 7 of the manuscript. I would rephrase "TBK1 controlling autophagy" as "TBK1 controlling anti-bacterial autophagy".

Done.

4. Two references are wrongly reported in the text in pages 6 and 8 of the manuscript.

Thank you for pointing out the error. Now corrected.

5. WIPI1 is erroneously spelled (WIPI-1), page 8 of the manuscript.

Corrected.

6. Figure 3 is erroneously reported as Figure 3B in the text.

Error has been corrected.

7. Fig. 4D. In order to help the reader, I would spell the acronyms (Hs, Bt, Gg, Xe) out in the figure legend. Out of curiosity, are these residues also conserved in *Mus musculus*?

*The acronyms have been spelt out in the figure legend as suggested.
Mice and rats do not encode full length NDP52.*

2nd Editorial Decision

13 May 2016

Thank you for submitting the revised version of your manuscript to us. It has now been seen by the original referees again, whose comments I enclose below.

As you will see, the referees appreciate that you added further insight, though it remains still unclear how TBK1 regulates WIPI1/2. Referee #2 and #3 are now broadly in favor of publication pending minor revision. I would thus like to invite you to provide a final version of your manuscript, addressing the remaining concerns. Most of the issues can be addressed in the discussion, but referee #2 still thinks that you should provide a negative control for the overexpression experiments.

REFeree REPORTS

Referee #1:

This is a substantially revised version of a previous manuscript that aims to investigate the mechanism by which TBK1 acts to promote anti-bacteria autophagy. The authors have added more data and have tried to address reviewer concerns.

The major difficulty with the manuscript is that there seem to be two disparate goals: (1) to identify how TBK1 mediates anti-bacterial autophagy and (2) how TBK1 is recruited to bacterial surfaces. This makes the data and discussion less focused.

The major new finding, which is of notable interest, is that TBK1 recruits the WIPI2 protein to bacterial surfaces. However, the paper does not clarify how the TBK1-WIPI2 axis regulates anti-bacterial autophagy, and WIPI2 was previously reported to participate in suppressing *Salmonella* cytosolic replication. The identification of WIPI2 as a downstream mediator for TBK1 is potentially very exciting, but without significant follow up, this part of the work is incomplete.

The bulk of the paper instead addresses the question of how TBK1 is recruited to bacteria surfaces. The authors show convincingly that multiple signals (galectin or ubiquitin) can serve to bring TBK1 to the bacteria surface, using a series of fusion proteins. This is an elegant approach, although the research is largely conducted in MEF cells, without follow up in relevant cell types. This part of the work is of interest, but represents a more incremental advance over previous knowledge.

Referee #2:

The authors put in a lot of efforts to determine the mechanism of actions of TBK1 kinase activity. The new data on OPTN and the WIPI family adds substance to the article and I'm generally satisfied with the new manuscript. The new data raises a lot of new questions, and unfortunately, the phosphorylation step mediated by TBK1 to recruit WIPI1/2 remains unknown, but I understand that this is now beyond the scope of the present article. However, I would like the authors to consider the following points from the previous round of revisions (points 2-4):

2) I thank the authors for pointing out that mice and rats do not have LC3C. But then, why indicate LC3C in Fig. 4C? It's misleading. Either remove it, or even better, indicate in the legend that LC3C is not found in mice and rats. Otherwise, I accept the authors' answer with regards to LC3 conversion.

3) Here I disagree; there are observations that seemingly contradict:

-In Verlhac et al. Fig. 1A, NDP52 restricts bacterial proliferation in HeLa cells, whereas the C425A mutant defective in Ub-binding does not.

-In this paper Fig. 5F, the fusion TBK1deltaC-NDP52-D439K is still able to restrict proliferation in TBK1-negative MEFs.

I agree with the authors that they may provide complementary insight, but there is a contradiction: based on the model presented by the present article, we can predict that expression of NDP52-D439K in a TBK1-positive MEFs will be able to restrict bacterial proliferation. NDP52-D439K binds TBK1, which ends up being the same as the fusion protein shown in Fig. 5F. How is that different from expressing NDP52-C425A in a NDP52-null background? There might be some differences between human and mouse cells that account for that. Moreover, the C425A mutant would unfold the ZnF domain, which can lead to aberrant protein-protein interactions. I don't know what the answer is, but I would only ask the authors to at least discuss the topic, as this is likely to raise questions from readers.

Also, it's not quite true that mice lack NDP52. Mice encode a homolog of NDP52, which lacks Ub- and Gal8-binding domains and indeed may not be "functional", i.e. it cannot play the same function as human NDP52 (it could have another function). But as the authors probably know, mice (and human too) have another adaptor referred to as TAX1BP1. This protein has SKICH, LIR, Coiled-coil and ZnF domains like NDP52, and binds ubiquitin. In any case, these are points to consider for the discussion about the importance of ubiquitin-binding in NDP52.

4) Here the authors have missed the point of my comment. Of course these adaptor protein constructs without TBK1 would not have kinase activity! But overexpression of these constructs could lead to the stimulation of some unknown pathway that triggers bacterial autophagy. This is the whole point about negative controls! We can predict they will be negative and they probably will turn out negative, but from the current data, there is no way to formally exclude the possibility that overexpression of these adaptors leads to bacterial autophagy independently of TBK1. The authors need to demonstrate that the effects observed are not caused by overexpression of these adaptors. I understand that these are not necessarily easy experiments to conduct, and I would be satisfied if the authors can point to experiments in the literature to support their case.

Referee #3:

The authors addressed most of my requests, either performing the experiments suggested or by using alternative approaches better fitting their experimental system.

In more details:

1) Despite the terrific efforts from the authors, the molecular mechanism by which TBK1 regulates the upstream regulators of autophagy (WIPI1/2) at the infection sites remains unclear (is it through a WIPI1/2 interactor? Could WIPI1/2 be direct targets of TBK1?). As I pointed out in the first revision, the link between TBK1-mediated selective autophagy and macroautophagy is exciting, and mechanistic insights would have further increased the relevance of the manuscript, making it more appealing for autophagy readers in general, rather than for people working on selective

autophagy. Nevertheless, this is an elegant and high-quality study, and the authors fulfilled all my other requests.

2) Altogether, as for my other remarks, the authors have put together an outstanding amount of convincing results, provided the reader with a full set of self-explanatory and complete figures, and corroborated their data with solid controls.

For this reason, in my opinion the present manuscript can be accepted for publication.
As a very last minor concern:

The GFP Western Blots in the Supplementary Figure 2A are saturated, and I suggest them to be replaced with less exposed and more reliable acquisitions.

2nd Revision - authors' response

23 May 2016

Referee #1:

This is a substantially revised version of a previous manuscript that aims to investigate the mechanism by which TBK1 acts to promote anti-bacteria autophagy. The authors have added more data and have tried to address reviewer concerns.

The major difficulty with the manuscript is that there seem to be two disparate goals: (1) to identify how TBK1 mediates anti-bacterial autophagy and (2) how TBK1 is recruited to bacterial surfaces. This makes the data and discussion less focused.

The major new finding, which is of notable interest, is that TBK1 recruits the WIPI2 protein to bacterial surfaces. However, the paper does not clarify how the TBK1-WIPI2 axis regulates anti-bacterial autophagy, and WIPI2 was previously reported to participate in suppressing Salmonella cytosolic replication. The identification of WIPI2 as a downstream mediator for TBK1 is potentially very exciting, but without significant follow up, this part of the work is incomplete.

The bulk of the paper instead addresses the question of how TBK1 is recruited to bacteria surfaces. The authors show convincingly that multiple signals (galectin or ubiquitin) can serve to bring TBK1 to the bacteria surface, using a series of fusion proteins. This is an elegant approach, although the research is largely conducted in MEF cells, without follow up in relevant cell types. This part of the work is of interest, but represents a more incremental advance over previous knowledge.

We thank referee 1 for his/her time and efforts in reviewing our manuscript.

Referee #2:

The authors put in a lot of efforts to determine the mechanism of actions of TBK1 kinase activity. The new data on OPTN and the WIPI family adds substance to the article and I'm generally satisfied with the new manuscript. The new data raises a lot of new questions, and unfortunately, the phosphorylation step mediated by TBK1 to recruit WIPI1/2 remains unknown, but I understand that this is now beyond the scope of the present article. However, I would like the authors to consider the following points from the previous round of revisions (points 2-4):

2) I thank the authors for pointing out that mice and rats do not have LC3C. But then, why indicate LC3C in Fig. 4C? It's misleading. Either remove it, or even better, indicate in the legend that LC3C is not found in mice and rats. Otherwise, I accept the authors' answer with regards to LC3 conversion.

We apologize for the confusion; Fig 4c is now changed.

3) Here I disagree; there are observations that seemingly contradict:
-In Verlhac et al. Fig. 1A, NDP52 restricts bacterial proliferation in HeLa cells, whereas the C425A mutant defective in Ub-binding does not.

-In this paper Fig. 5F, the fusion TBK1deltaC-NDP52-D439K is still able to restrict proliferation in TBK1-negative MEFs.

I agree with the authors that they may provide complementary insight, but there is a contradiction: based on the model presented by the present article, we can predict that expression of NDP52-D439K in a TBK1-positive MEFs will be able to restrict bacterial proliferation. NDP52-D439K binds TBK1, which ends up being the same as the fusion protein shown in Fig. 5F. How is that different from expressing NDP52-C425A in a NDP52-null background? NDP52 C425A is therefore deficient in at least one of its potentially many functions. However, as shown in Fig 5F, recruitment of TBK1 to cytosol-invading bacteria is fully functional. One must therefore conclude that Verlhac studied a function of NDP52 different to its role in TBK1 recruitment.

We completely agree with the referee's chain of thought. Verlhac studied a function of NDP52 distinct to its role in TBK1 recruitment, namely, as indicated in Verlhac's title, the maturation of pathogen-containing autophagosomes.

There might be some differences between human and mouse cells that account for that. Moreover, the C425A mutant would unfold the ZnF domain, which can lead to aberrant protein-protein interactions. I don't know what the answer is, but I would only ask the authors to at least discuss the topic, as this is likely to raise questions from readers.

While the C425A mutant used by Verlhac is certainly not ideal from a structural point of view, the basic conclusion remains unchanged, namely that Verlhac studied an allele deficient in autophagosome maturation, while NDP52 D439K specifically lacks ubiquitin binding but still recruits TBK1 via galectin8.

Also, it's not quite true that mice lack NDP52. Mice encode a homolog of NDP52, which lacks Ub- and Gal8-binding domains and indeed may not be "functional", i.e. it cannot play the same function as human NDP52 (it could have another function).

The referee is correct that the first 130 triplets or so of NDP52 are present in the murine genome. However, expression databases suggest that it is not expressed in any tissue except ES cells (<http://biogps.org/#goto=genereport&id=76815>).

But as the authors probably know, mice (and human too) have another adaptor referred to as TAX1BP1. This protein has SKICH, LIR, Coiled-coil and ZnF domains like NDP52, and binds ubiquitin. In any case, these are points to consider for the discussion about the importance of ubiquitin-binding in NDP52.

We agree with the referee on a potential function of TAX1BP1, particularly in murine cells. We therefore performed preliminary experiments with TBK1-deltaC-Tax1BP1 fusion constructs but did not obtain data that would add mechanistic insight beyond what is provided in the manuscript.

4) Here the authors have missed the point of my comment. Of course these adaptor protein constructs without TBK1 would not have kinase activity! But overexpression of these constructs could lead to the stimulation of some unknown pathway that triggers bacterial autophagy. This is the whole point about negative controls! We can predict they will be negative and they probably will turn out negative, but from the current data, there is no way to formally exclude the possibility that overexpression of these adaptors leads to bacterial autophagy independently of TBK1. The authors need to demonstrate that the effects observed are not caused by overexpression of these adaptors. I understand that these are not necessarily easy experiments to conduct, and I would be satisfied if the authors can point to experiments in the literature to support their case.

We appreciate that the referee invites us to provide support from the literature. However, since we are not aware of any published overexpression studies in TBK1 deficient cells, we will argue the case based on our experimental data and first principles.

The referee suggests that overexpression of TBK1deltaC fused to seven different genes (Nap1, NDP52, Galectin-8, Optineurin, Nemo, Tab2, Mud1) may restrict bacterial proliferation due to an

unknown pathway, and not because these constructs recruit TBK1 to bacteria. The referee's pathway would need to be active in TBK1^{-/-} cells, which, as we have shown, fail to recruit the essential autophagy mediator WIPI2 and hence do not execute anti-bacterial autophagy. We consider the scenario proposed by the reviewer very unlikely for the following reasons:

1.) As reported in our previous point-to-point reply, at least one of the seven genes (*Optn*)

does not provide anti-bacterial activity if overexpressed in TBK1^{-/-} cells. For your reference, I copy those data below.

- 2.) The seven overexpressed proteins are not related to each other, nor do they contain any homologous domains or share any biochemical activity. A single (unknown) effector mechanism, as suggested by the reviewer, is therefore mechanistically difficult to envision. The alternative possibility of multiple (unknown) effector mechanisms (one per gene) is not parsimonious and hence even less likely.
- 3.) Some of the above proteins were truncated to single domains (for example Nap1 to only 85aa). Our interpretation of the results is that these domains, if fused to TBK1 Δ C, mediate anti-bacterial autophagy by recruiting TBK1 to bacteria. The alternative mechanism suggested by the reviewer would imply that each of these completely unrelated single domains provides an anti-bacterial effector function. Considering their small size, we find such a scenario even more unlikely than for full-length proteins.

- 4.) *In contrast, our hypothesis of TBK1 recruitment driving anti-bacterial autophagy enabled us to correctly predict proteins that, upon fusion to TBK1deltaC, mediate anti-bacterial autophagy. Note that the predictive power of our hypothesis expanded even to yeast proteins, a species not known to execute anti-bacterial autophagy.*

Taken together, we provide direct evidence for at least one of the seven proteins (Optineurin) that they restrict bacterial proliferation because they recruit TBK1 to invading bacteria and not because they harbor an unknown effector activity. We consider the possibility that overexpression of the remaining six proteins would trigger anti-bacterial autophagy independently of TBK1 very unlikely. However, we do agree with the referee that a TBK1-independent overexpression phenomenon remains a formal possibility, which in an ideal world should be investigated experimentally. From a practical point of view, however, we wonder whether such experiments are justifiable, considering the time and resources they would require. We estimate conservatively that the de-novo generation of additional stable cell lines overexpressing Nap1, NDP52, galectin8, optineurin, Nemo, Tab2, and Mud1, followed by infection experiments would require about two months and significant funds for reagents and salary. We would therefore appreciate editorial guidance, whether these experiments are essential for the acceptance of our manuscript.

Referee #3:

The authors addressed most of my requests, either performing the experiments suggested or by using alternative approaches better fitting their experimental system.

In more details:

- 1) Despite the terrific efforts from the authors, the molecular mechanism by which TBK1 regulates the upstream regulators of autophagy (WIPI1/2) at the infection sites remains unclear (is it through a WIPI1/2 interactor? Could WIPI1/2 be direct targets of TBK1?). As I pointed out in the first revision, the link between TBK1-mediated selective autophagy and macroautophagy is exciting, and mechanistic insights would have further increased the relevance of the manuscript, making it more appealing for autophagy readers in general, rather than for people working on selective autophagy. Nevertheless, this is an elegant and high-quality study, and the authors fulfilled all my other requests.
- 2) Altogether, as for my other remarks, the authors have put together an outstanding amount of convincing results, provided the reader with a full set of self-explanatory and complete figures, and corroborated their data with solid controls.

For this reason, in my opinion the present manuscript can be accepted for publication.

We thank referee 3 for his/her efforts in reviewing our manuscript.

As a very last minor concern:

The GFP Western Blots in the Supplementary Figure 2A are saturated, and I suggest them to be replaced with less exposed and more reliable acquisitions.

The blots have been replaced.

3rd Editorial Decision

24 May 2016

Thank you very much for sending the revised version of your manuscript. I appreciate the amendments and the response to the referees' concerns, and I am happy to accept your manuscript for publication in The EMBO Journal.

Corresponding Author Name: felix randow

Journal Submitted to: EMBO J

Manuscript Number: EMBOJ-2016-94491